# Moisture origin as a driver of temporal variabilities of the water vapour isotopic composition in the Lena River Delta, Siberia

Jean-Louis Bonne[1], Hanno Meyer[2], Melanie Behrens[1], Julia Boike[2,3], Sepp Kipfstuhl[1], Benjamin Rabe[1], Toni Schmidt[2], Lutz Schönicke[2], Hans Christian Steen-Larsen[4], Martin Werner[1]

[1]Alfred-Wegener-Institut Helmholtz-Zentrum für Polar- und Meeresforschung, Bremerhaven, 27515 Germany
[2]Alfred-Wegener-Institut Helmholtz-Zentrum für Polar- und Meeresforschung, Potsdam, 14401, Germany
[3]Geography Department, Humboldt-Universität zu Berlin, Berlin, 10099, Germany
[4]Geophysical Institute, University of Bergen, and Bjerknes Centre for Climate Research, Bergen, 5020, Norway

*Correspondence to*: Jean-Louis Bonne (jean-louis.bonne@awi.de), Martin Werner (martin.werner@awi.de)

**Abstract.** In the context of the Arctic amplification of climate change affecting the regional atmospheric hydrological cycle, it is crucial to characterize the present-day's moisture sources of the Arctic. The isotopic composition is an important tool to enhance our understanding of the drivers of the hydrological cycle, due to the different molecular characteristics of water stable isotopes during phase change. This study introduces two years of continuous in situ water vapour and precipitation isotopic observations conducted since July 2015 in the east-Siberian Lena delta, at the research station on the Samoylov Island. The vapour isotopic signals are dominated by variations at seasonal and synoptic time scales. Diurnal variations of the vapour isotopic signals are masked by synoptic variations, indicating low variations of the amplitude of local sources at the diurnal scale in winter, summer and autumn. Low amplitude diurnal variations in spring may indicate exchange of moisture between the atmosphere and the snow-covered surface. Moisture sources diagnostics based on semi-Lagrangian backward trajectories reveal that different air mass origins have contrasted contributions to the moisture budget of the Lena delta region. At the seasonal scale, the distance from the net moisture sources to the arrival site strongly varies. During the coldest months, no contribution from local secondary evaporation is observed. Variations of the vapour isotopic composition during the cold season on synoptic time scale are strongly related to moisture source regions and variations in the atmospheric transport: warm and isotopically-enriched moist air is linked with fast transport from the Atlantic sector; while dry and cold air with isotopically-depleted moisture is generally associated to air masses moving slowly over northern Eurasia.

## 1 Introduction

The amplitude of climate change in Arctic regions is likely to affect the atmospheric hydrological cycle, as sea ice is retreating and the temperatures are increasing, modifying the sources of evaporation and the saturation vapour pressure of the atmosphere. However, changes in evapotranspiration of Arctic regions during recent decades are poorly known, so far (Vihma et al., 2015)(Vihma et al., 2015).

Isotopic concentrations are commonly reported as $\delta^{18}O$ and $\delta^{2}H$, representing for the relative abundances of $H_2^{18}O$ and $H^2H^{16}O$, respectively, compared to the most abundant isotopologue $H_2^{16}O$. As isotope fractionation occurs during phase changes of water, they are largely used as tracers of the hydrological processes in the atmosphere and can be interpreted as proxies for past temperature variations in different types of climatic archives, such as ice cores, speleothems or ice wedges, specific ground ice features found in permafrost environments.

The $\delta^{18}O$ and $\delta^{2}H$ of precipitation and vapour at the global scale are primarily determined by the cooling-induced equilibrium distillation of the moisture from the source towards the measuring site. To the first order, this distillation affects both $H_2^{18}O$ and $H^2H^{16}O$ similarly, causing a global linear distribution of $\delta^{2}H$ versus $\delta^{18}O$ with a slope close to 8 (Craig, 1961)(Craig, 1961). Additional phase change processes occurring during non-equilibrium, namely kinetic fractionation processes, such as fast evaporation from the ocean surface or snow crystals formation, are visible in deviations from this relationship (Dansgaard, 1964; Jouzel and Merlivat, 1984; Merlivat and Jouzel, 1979)(Dansgaard, 1964; Jouzel and Merlivat, 1984; Merlivat and Jouzel, 1979). To study the impacts of kinetic fractionation, the second order parameter deuterium excess, hereafter d-excess, has been defined as the deviation from this $\delta^{2}H$ to $\delta^{18}O$ relationship (Dansgaard, 1964)(Dansgaard, 1964).

Present-day observations of vapour and precipitation events refine the understanding of various phases of the atmospheric water cycle and their imprint on the water isotopic compositions (Galewsky et al., 2016)(Galewsky et al., 2016). Such studies focus on exchange processes between the atmosphere and the Earth's surface, over open oceans (Benetti et al., 2014, 2017; Bonne et al., 2019; Zannoni et al., 2019)(Benetti et al., 2014, 2017; Bonne et al., 2019; Zannoni et al., 2019), ice sheets (Madsen et al., 2019; Steen-Larsen et al., 2014)(Madsen et al., 2019; Steen-Larsen et al., 2014), sea ice surfaces (Bonne et al., 2019)(Bonne et al., 2019), or continents (Bastrikov et al., 2014)(Bastrikov et al., 2014). Other studies focus on the atmospheric transport of moisture and show that typical vapour isotopic signals can be associated with distinct patterns of moisture origins (Bonne et al., 2014, 2015; Guilpart et al., 2017; Kopec et al., 2014; Steen-Larsen et al., 2013, 2015)(Bonne et al., 2014, 2015; Guilpart et al., 2017; Kopec et al., 2014; Steen-Larsen et al., 2013, 2015). However, the systematic relationship between water isotopes and atmospheric moisture transport remains uncertain in many conditions and locations. Recently, water vapour isotopic observations have also been showing great potential when used to benchmark the simulations of the hydrological cycle in General Circulation Models equipped with water isotopes (Steen-Larsen et al., 2017)(Steen-Larsen et al., 2017).

In the eastern Arctic region, water isotopic measurements of precipitation samples have been conducted at a land station along the Lena river (Zhigansk, 66.8°N, 123.4°E, 92 m above sea level) during multiple years (2004-2008) and have been combined with ship-based water vapour isotopic observations based on discrete samples during a campaign in the eastern Arctic Ocean on a period extending from before the sea-ice minimum to the beginning of the sea-ice growth season (Kurita, 2011)(Kurita, 2011). This study revealed that higher values of d-excess where observed at the land station in mid-Autumn

for air masses originating from the Arctic Ocean region, compared to air masses originating from lower latitudes. A hypothesis to explain these high d-excess values was that evaporation over the ocean was taking place during the sea-ice growth season at very low relative humidity: the dry air masses originating from sea ice covered areas enhance evaporation with strong kinetic fractionation when passing over the open ocean.

More recently, continuous water vapour isotopic observations have been conducted in western Russia, at the Kourovka observatory (Bastrikov et al., 2014; Gribanov et al., 2014)(Bastrikov et al., 2014; Gribanov et al., 2014) and at Labytnangy (Konstantin Gribanov and Jean Jouzel, personal communication). Observations from the Kourovka station depicted seasonal and synoptic variations as well as strong diurnal variations in summer on the first and second order vapour isotopic signals. To explain the lack of relationship between vapour isotopic signal and meteorological parameters (humidity and temperature) in summer, a strong contribution of continental recycling and local evapotranspiration has been suggested (Butzin et al., 2014)(Butzin et al., 2014). These observations were also used to test the ability of an isotope-enabled Atmosphere General Circulation Model to reproduce the water vapour isotopic composition and have shown an excellent correlation between simulation and water vapour $\delta^2H$ values measured at the surface (Gribanov et al., 2014)(Gribanov et al., 2014).

In this study, we focus on water vapour isotopic observations continuously performed from July 2015 to June 2017 at the research station on Samoylov Island in Lena delta (72°22' N, 126°29' E), in the Russian Arctic. This station is at higher latitude and much further east than the Kourovka and Labytnangy stations and will therefore be more representative of an Arctic continental climate. We assess the variations of water vapour isotopic composition at different time scales, from the seasonal to the synoptic and diurnal scales, and relate them with the variations of local meteorological parameters. In order to interpret these variations in a larger regional context, we use atmospheric transport simulations together with moisture source diagnostics. These simulations help identify the main moisture advection regimes of the region and decipher the imprint of local versus remote moisture sources on the locally observed water vapour isotopic composition.

## 2 Method

### 2.1 Study area

The observations presented in this study have been conducted , in the Lena river delta, north-east Siberia (Figure 1 a), at the research station on the Samoylov Island (72°22'N, 126°29'E) (Figure 1 b,c). The Lena river flows from its source in the Baikal Mountains towards north into the Laptev Sea where it forms a large delta of 150 km in diameter. In its northern part, the river flows alongside the Verkhoyansk Mountains on its eastern side (see Figure 1).  The Samoylov island consists on its western part of the modern flood plain and on its eastern part of a Holocene terrace characterised by polygonal tundra and larger water bodies (see Figure 1 c and Supplementary Figure 1). With a mean annual temperature below -12 °C, minimum

winter air temperatures below -45 °C and summer air temperatures that can exceed 25 °C, the region has a typical Arctic continental climate (Boike et al., 2013, 2019a)~~(Boike et al., 2013, 2019a)~~. Average annual rainfall is about 169 mm. The

95 winter snow cover is thin (~0.3 m), with a maximum recorded of about 0.8 m in 2017. The snow accumulation starts in late September and all the snow cover completely melts over a few days in early June. Permafrost underlays the study area to depths between 400 and 600 m (Grigoriev, 1960)~~(Grigoriev, 1960)~~, and the rate of permafrost temperature warming at the zero annual amplitude level (ZAA) is one of the highest recorded (Biskaborn et al., 2019)~~(Biskaborn et al., 2019)~~. The active layer thawing period starts at the end of May and the active layer thickness reaches a maximum at the end of

100 August/beginning of September.

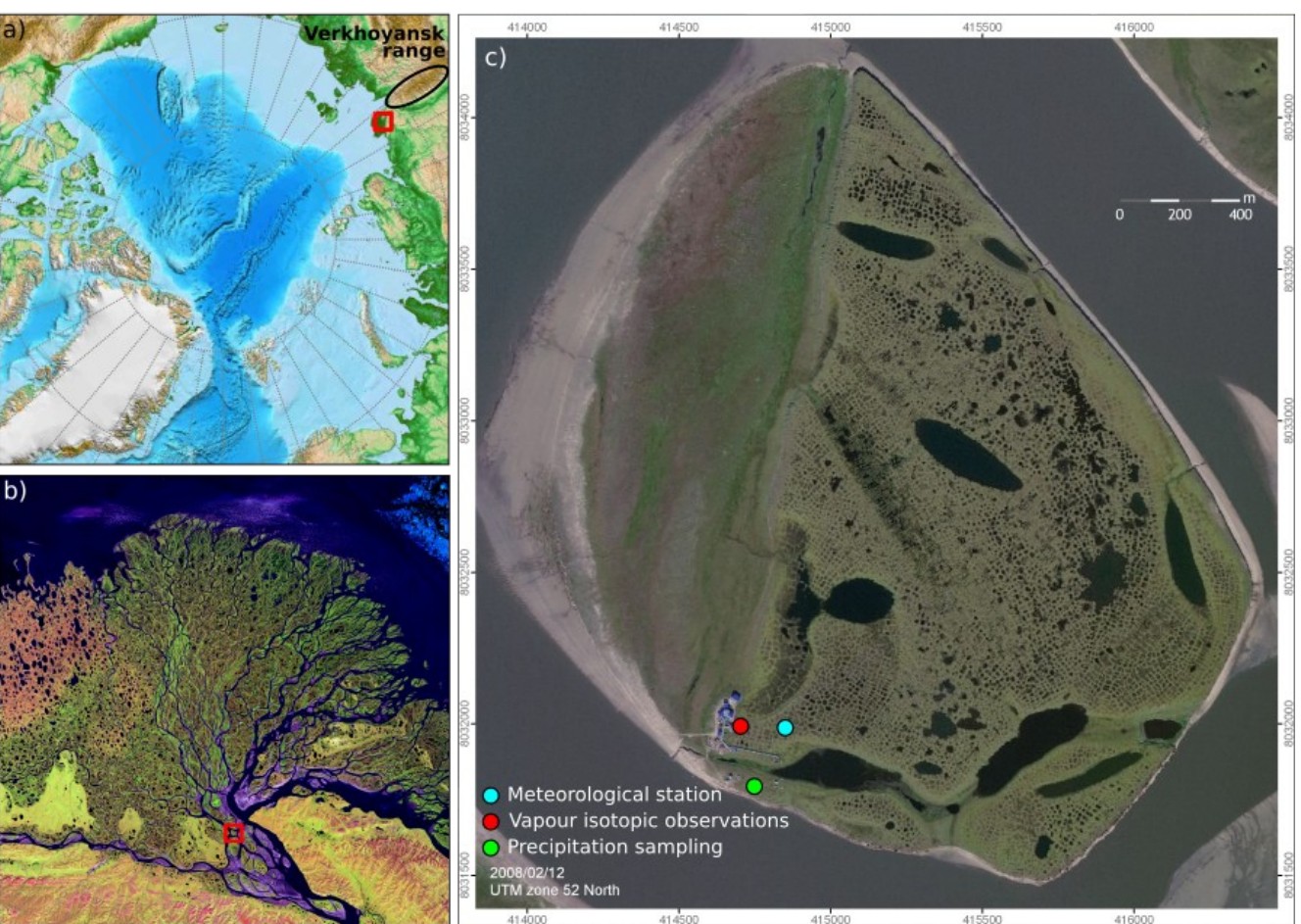

**Figure 1: Maps of the study area: (a) In the Arctic region (North pole Lambert azimuthal projection). The Verkhoyansk range is indicated with a black ellipse. (b) Within the Lena River Delta, Eastern Siberia (Landsat 7, 2000; image provided by the USGS EROS Data Center Satellite Systems Branch). (c) Map of the Samoylov island**

**(Source: Esri, DigitalGlobe, GeoEye, Earthstar Geographics, CNES/Airbus DS, USDA, USGS, AeroGRID, IGN, and the GIS User Community), with locations of the meteorological station, water vapour isotopic observations and precipitation sampling set-up respectively displayed as blue, red and green dots.**

## 2.2 Meteorological observations

We used meteorological records from the Samoylov Island available from the PANGAEA database (Boike et al., 2019b) (Boike et al., 2019b), conducted as part of a study focusing on the influence of meteorological parameters on the local permafrost evolution (Boike et al., 2019a)(Boike et al., 2019a). The location of the meteorological observations is displayed on Figure 1 c. The temporal resolution of this dataset in our period of interest is 30 minutes. The meteorological parameters were measured at an approximate distance of 200 m eastward from our water isotopic observations. Here, we use the following parameters, always measured above the snow cover during the periods where snow is present: relative humidity (RH, expressed in %) and air temperature (expressed in °C), measured at 2 m above ground level; wind speed and wind direction (expressed in m s-1 and °), measured at 3 m above ground level.

## 2.3 $\delta$-notation for water isotopic compositions and d-excess

Isotopic compositions of samples are expressed as $\delta^{18}O$ and $\delta^2H$ in permille unit (‰). $\delta$ values are defined as (Craig, 1961) (Craig, 1961):

$$\delta = 1000 \cdot \left( \frac{R_{sample}}{R_{VSMOW}} - 1 \right), \tag{1}$$

where $\delta$ can be either $\delta^{18}O$ or $\delta^2H$, with $R_{sample}$ and $R_{VSMOW}$ as the isotopic ratios ($H_2^{18}O/H_2^{16}O$ or $H^2H^{16}O/H_2^{16}O$, respectively) of the sample and the Vienna-Standard Mean Ocean Water (VSMOW) reference (Coplen, 2011)(Coplen, 2011).

We calculated the deuterium excess, noted d-excess using its classical definition (Dansgaard, 1964)(Dansgaard, 1964):

$$\text{d-excess} = \delta^2H - 8 \cdot \delta^{18}O \tag{2}$$

## 2.4 Water vapour isotopic observations

A Cavity Ring Down Spectrometer (CRDS) was installed in July 2015 at the Samoylov station in order to continuously record the near surface water vapour isotopic composition at an approximate 1 Hz frequency. Its location is displayed on Figure 1 c. The analyser is placed inside a heated container (an electric heater placed inside the container avoids the temperature to drop below the freezing point in winter) and the ambient air inlet is located at 5 m above the ground level, above the roof of the container. The inlet tube (1/4 inches diameter stainless steel tube of approximately 4 m length) is

insulated and constantly heated by a heating wire at around 50 °C, independently from the container heating. Following the recommendations for long-term calibration of CRDS water vapour isotopic analysers (Bailey et al., 2015)(Bailey et al., 2015), a custom-made calibration system allows the automatic correction of (i) the concentration-dependence of the isotopic measurements as a function of the humidity level and (ii) the deviation from the Vienna Standard Mean Ocean Water (VSMOW2) (Coplen, 2011) (Coplen, 2011) scale on a daily basis. The system includes a custom-made vaporizer system (as described in Bonne et al., 2019)(as described in Bonne et al., 2019) allowing the measurement of four different water isotopic standards, and a bubbler system (similar to the one described in Steen-Larsen et al., 2014) (similar to the one described in Steen-Larsen et al., 2014) with one standard water of known isotopic composition (see schematics of the system in Supplementary Figure 2). The data treatment and calibration procedures are similar as for Bonne et al. (2019)Bonne et al. (2019). Details on the calibration applied and the stability of the standards measurements are given in the Supplementary Note 1. We note $\delta^{18}O_v$, $\delta^2H_v$ and d-excess$_v$ the isotopic values of the water vapour. Based on the uncertainty of both corrections of the humidity-concentration dependence and deviations from the VSMOW-SLAP scale, the measurement accuracy is estimated at 0.6 ‰, 3.0 ‰ and 5.7 ‰ on $\delta^{18}O_v$, $\delta^2H_v$ and d-excess$_v$. The precision is estimated from the standard deviation of calibration standard measurements at a constant humidity level. For values averaged at a 1-h resolution,. the precision measurements performed under humidity levels higher than 3 g/kg is equal to 0.25 ‰, 0.5 ‰ and 2 ‰ on $\delta^{18}O_v$, $\delta^2H_v$ and d-excess$_v$. This precision deteriorates logarithmically for drier air conditions, reaching 2.0 ‰, 6.5 ‰ and 17 ‰ for $\delta^{18}O_v$, $\delta^2H_v$ and d-excess$_v$ for humidity levels of 0.3 g/kg. The dataset presented in this study has been averaged at a 6-h temporal resolution (except for the calculation of the diurnal cycle based on the hourly averaged dataset) and the precision can therefore be estimated at 0.10 ‰, 0.2 ‰ and 1 ‰ on $\delta^{18}O_v$, $\delta^2H_v$ and d-excess$_v$ for humidity levels above 3 g/kg and 0.8 ‰, 2.7 ‰ and 7 ‰ for $\delta^{18}O_v$, $\delta^2H_v$ and d-excess$_v$, for humidity levels of 0.3 g/kg.

Technical difficulties linked with the extreme cold and dry conditions during winter and the remote location of the station lead to some periods of missing data. Modifications of the instrumental setup and calibration procedure during the successive maintenance expeditions allowed to progressively enhance the precision of the measurements.

## 2.5 Water isotopic composition of event-based precipitation samples and calculation of equilibrium vapour isotopic composition

Precipitation sampling is carried out after each rain and snowfall event using a funnel construction tightly connected (with a rubber plug to avoid evaporation) to a 250 ml HDPE bottle, which is emptied after each precipitation event. The 250ml bottle is placed inside a larger tube to protect it from wind. The funnel is connected to this larger tube with tent cords and fixed to the ground as well. The set-up is located at about 250 m from the location of the water vapour isotopic observations as displayed on Figure 1 c.

Event-based precipitation samples retrieved over the period of the study have been measured for their water isotopic composition in the stable isotope laboratory of the Alfred Wegener Institute, Helmholtz Centre for Polar and Marine

Research, in Potsdam. Isotope ratios were determined by isotope ratio mass spectrometry using a Finnigan MAT Delta-S mass spectrometer, applying the equilibration technique (Meyer et al., 2000)(Meyer et al., 2000). The measurement accuracy is better than 0.1 ‰ for $\delta^{18}O_p$ and 0.8 ‰ for $\delta^2H_p$ (Meyer et al., 2000)(Meyer et al., 2000). We note $\delta^{18}O_p$, $\delta^2H_p$ and d-excess$_p$ the isotopic values of precipitation.

To compare the vapour and precipitation isotopic dataset, we calculate the isotopic composition of a theoretical water vapour at equilibrium with the precipitation, from the measured $\delta^{18}O_p$ and $\delta^2H_p$ values. We use the measured 2 m air temperature to estimate the value of the $\alpha_{eq}$ equilibrium fractionation coefficients between liquid (solid) and vapour for air temperatures above (below) the freezing point (Majoube, 1971a, 1971b; Merlivat and Nief, 1967)(Majoube, 1971a, 1971b; Merlivat and Nief, 1967). For each precipitation sample, with an isotopic ratio denoted R$_P$, the theoretical isotopic ratio of vapour, denoted R$_V$, is given by R$_V$ = R$_P$/$\alpha_{eq}$.

## 2.6 Selection of synoptic extrema from meteorological and vapour isotopic parameters

We identify the extrema occurring at a synoptic time scale for a set of parameters based on the daily averaged dataset: temperature, logarithm of the specific humidity, $\delta^{18}O_v$ and d-excess$_v$. We use the logarithm of specific humidity for comparability regarding its logarithmic relationship with temperature (Clausius-Clapeyron relationship). For each parameter, we compute the distances of the daily values to a 60-days running average and identify the extrema as the first and last deciles of this dataset (low and high extrema, respectively). Some data gaps exist at different periods for the temperature, specific humidity and the isotopic parameters. In order to have a comparable selection of extrema for all parameter, without influence of the different data gaps between the datasets, we base this selection of synoptic time scale extrema only on data with simultaneously valid measurements for all parameters. No water vapour isotopic data are removed due to this selection, but it  leads to the filtering of respectively 3.7 % and 13.1 % of the specific humidity and temperature data.

## 2.7 Sea ice cover data

The sea ice coverage within the 500 km area surrounding the station has been derived from ERA-interim reanalyses (Dee et al., 2011)(Dee et al., 2011) at 0.75° × 0.75° spatial and 6-h temporal resolution. Results are shown as a proportion of sea ice covered surface compared to the surface of the surrounding 500 km area. As this area includes grid cells covered by land, the maximum value is lower than 1.

## 2.8 Moisture source diagnostics

The origin of the moisture arriving at our research area is estimated using a moisture source diagnostics method (Sodemann et al., 2008)(Sodemann et al., 2008) based on semi-Lagrangian simulations of air masses transport with the model

FLEXPART version 8.1 (Stohl et al., 2005)(Stohl et al., 2005). Meteorological fields from the ECMWF ERA-interim dataset (Dee et al., 2011)(Dee et al., 2011) at 0.5° horizontal resolution and 137 vertical levels are used. Air parcels are traced 10 days backward in time from a box centred around the Samoylov station. The box has a 3°x6° latitudinal and longitudinal width and spans altitudes between 0 and 500 m. All simulation outputs are presented over a global grid of 1°x1° resolution representing the summed contribution of all individual particles. From the air mass trajectory simulations, without considering the transport of moisture yet, we calculate the parameter "trajectories locations", as the summed number of air masses passing within each grid cell over all time steps. This parameter is representative of the location of transported air masses.

As different air masses can carry various amounts of moisture, a complementary calculation is conducted to properly evaluate the atmospheric transport of moisture: this moisture source diagnostics consists of interpreting the increase or decrease of the moisture content of air masses between different successive time steps along their trajectories respectively as an input or output of moisture contributing to the total moisture at the end of the trajectories. The contribution of every single air mass in terms of input and output of moisture is summed over all model output grid cells. Different parameters are estimated over this grid. The parameter "moisture uptake" (expressed in mm/day) represents the amount of moisture injected to the air masses within each grid cell. Moisture uptakes are interpreted as evaporation, transpiration or sublimation at the surface when the air masses are below an altitude equal to 1.5 times the boundary layer height. Moisture uptakes are interpreted as the result of mixing of different air masses or re-evaporation of falling precipitation when the air masses are above this threshold, therefore within the free troposphere (Dütsch et al., 2017)(Dütsch et al., 2017). Therefore, the parameters "boundary layer moisture uptake" and "free troposphere moisture uptake" are presented separately. The parameter "evaporation minus precipitation" (expressed in mm/day) represents the sum of the differences between the moisture uptakes and moisture losses of all particles occurring in each grid cell.

## 3 Results

### 3.1 Ranges of variations of isotopic and meteorological dataset

Over the observational period (2015-07-01 to 2017-07-01), the dataset comprises a total of 2395 water vapour isotopic values, at a 6-hours resolution. All observed parameters, presented in Figure 2, are characterized by variations at the inter-annual, seasonal, synoptic and eventually diurnal timescales.

For temperature and specific humidity distributions, the amplitude of seasonal variations is larger than the amplitude of synoptic variations (Figure 2). The complete observation period presents averages of specific humidity and temperature of 2.4 g kg$^{-1}$ and -11 °C (Table 1). Minimal and maximal temperatures reach -41.1 °C and +21.5 °C, respectively (Table 1). This two-years period is not exceptional, as observations over 16-years on this site have recorded temperature extremes below -45 °C and above +25 °C, with an average mean annual temperature of -12 °C (Boike et al., 2019a)(Boike et al.,

2019a), close to the average on our observational period. Specific humidity values reach extremely dry values in winter, down to 0.06 g kg$^{-1}$, and maximum values in summer up to 12 g kg$^{-1}$. A non-linear distribution between air temperature and specific humidity is observed (Figure 3 b), as expected from the Clausius-Clapeyron relationship. Relative humidity values vary between 40 % and 102 %, with a mean value of 84 % (Table 1).

Important local hydrological changes are expected between evapotranspiration or sublimation-dominated regimes during the warm and the cold seasons. The surroundings of the station are indeed characterised by contrasted surface covers between these seasons, with ice-free or ice-covered Lena river, land areas covered by vegetation or snow and the surrounding Laptev Sea covered by sea ice during the cold season. Locally measured snow depth indicates the continuous presence of a snow cover from the last days of September to June. Melting takes place within a few days, until mid-June (in 2016) or end of June (in 2017), as seen on Figure 2. Satellite images from the MODIS sensor on the Terra satellite reveal that ice covering the river in the Lena delta completely disappears within a few days, between May 23$^{rd}$ and June 10$^{th}$ in 2016 and between May 29$^{th}$ and June 13$^{th}$ in 2017. The freeze-up of the Lena river also takes place within a few days and has been estimated between October 10$^{h}$ and 20$^{th}$ in 2015 and between October 15$^{th}$ and 22$^{nd}$ in 2016. The sea ice cover on the Laptev sea, as estimated from the proportion of sea ice within the surrounding 500 km derived from the ERA-interim data, also depicts the fast opening of a polynya in early June (Figure 2). Presence of a significant sea ice cover in the region is still observed in July and the complete disappearance of sea ice occurs by the end of August. The complete sea ice cover then builds up within a few weeks in October (in mid-October in 2015 and in the second half of October in 2016). Many different surface cover changes take place simultaneously or within a few days or weeks in autumn and spring. There is no simple situation without snow but with a complete sea ice and river ice, or the opposite situation. It is therefore difficult to investigate the impacts of the evolution of each potential local moisture source individually.

For $\delta^{18}O_v$ and $\delta^{2}H_v$, some episodes of winter synoptic variations reach almost the amplitude of the seasonal cycle, while the d-excess$_v$ signal is predominated by synoptic variations (Figure 2). The vapour $\delta^{18}O_v$, $\delta^{2}H_v$ and d-excess$_v$ have averaged values over the complete dataset of respectively -36.9 ‰, -270 ‰ and 24.6 ‰ (Table 1). Water vapour isotopic composition minima and maxima are -53.2 ‰ and -20.9 ‰ for $\delta^{18}O_v$, -382 ‰ and -155 ‰ for $\delta^{2}H_v$ and 3 ‰ and +62 ‰ for d-excess$_v$. At Samoylov, the $\delta^{18}O_v$ and $\delta^{2}H_v$ are overall strongly correlated, with the following empirical relationship: $\delta^{2}H_v = 7.2\ \delta^{18}O_v - 6.7$, $R^2 = 0.98$ (Figure 3 a). This slope is a little lower than the slope of 7.5 obtained from observations at the Kourovka observatory in western Siberia (Bastrikov et al., 2014)(Bastrikov et al., 2014) and lower than the LMWL of Samoylov precipitation (LMWL: $\delta^{2}H_p = 7.6\ \delta^{18}O_p - 5.5$, n = 208, $R^2 = 0.95$, p < 0.05). While the warm seasons isotopic compositions are similar between both years, significant inter-annual differences between the average d-excess$_v$ values during the cold seasons are noticed, with higher d-excess$_v$ values during winter and spring 2016 than in 2017 (average and standard deviations on d-excess$_v$ of +32.5 ‰ +/- 9.0 ‰ from December 2015 to April 2016 and of +22.7 ‰ +/- 7.5 ‰ from December 2016 to April 2017).

The precipitation samples also depict inter-annual, seasonal and synoptic time scale variations. Altogether, precipitation samples depict more enriched δ-values and lower d-excess than vapour, with mean values over the complete observation

period for $\delta^{18}O_p$, $\delta^2H_p$ and d-excess$_p$ of respectively -21.6 ‰, -169.3 ‰ and 3 ‰ (Table 1). The difference between vapour and precipitation is stable throughout all seasons for $\delta$-values, but is varying between seasons for the d-excess with higher gaps in spring (maximum gap) and winter than in summer (minimum gap) and autumn. Precipitation samples have on average 14.5 ‰ lower d-excess values than vapour in summer and 28.5 ‰ lower d-excess values in spring, as shown in

Table 1. Significant inter-annual variations are noticed during the cold seasons on the values of $\delta^{18}Op$ and $\delta^2H_p$ and particularly on d-excess$_p$, with higher $\delta^{18}Op$ and $\delta^2H_p$ and lower d-excess$_p$ values during winter and spring 2016 than in 2017 (average and standard deviations on $\delta^{18}O_p$ and d-excess$_p$ respectively of -24.7 ‰ +/- 5.4 ‰ and -9.1 +/- 17.8 ‰ from December 2015 to April 2016 and of -29.2 ‰ +/- 5.2 ‰ and +9.5 ‰ +/- 7.7 ‰ from December 2016 to April 2017). The difference in precipitation d-excess$_p$ levels between both years is in the opposite direction compared to the difference

observed in the vapour d-excess$_v$.

The minima of specific humidity are associated with the lowest $\delta^{18}O_v$ and $\delta^2H_v$ and the highest d-excess$_v$ values, comparable to vapour observations performed on the East Antarctic Plateau in summer (Casado et al., 2016; Ritter et al., 2016)(Casado et al., 2016; Ritter et al., 2016). This could indicate a strong isotopic depletion during long-range atmospheric transport from the moisture sources to the location of the observations, or recycling of moisture with sublimation over the surrounding

snow-covered areas (Pang et al., 2019)(Pang et al., 2019). Such high d-excess$_v$ values do not only reflect kinetic fractionation processes, but are also partly due to the variations of the $\delta^{18}O_v$ to $\delta^2H_v$ relationship during equilibrium fractionation at very low temperatures (Dütsch et al., 2017)(Dütsch et al., 2017).

There is a linear relationship between $\delta^{18}O_v$ and air temperature (with a slope of 0.44 ‰ °C$^{-1}$, $R^2$=0.75, p< 0.01, Figure 3 c), but with a large scatter clearly demonstrating that the air temperature variations do not fully explain the $\delta^{18}O_v$ variations.

This linear relationship observed in vapour is close to the 0.4 ‰ °C$^{-1}$ relationship obtained both for local precipitation $\delta^{18}O_p$ on event and monthly means, as well as for $\delta^{18}O_v$ and T in the west Siberian Kourovka station. A non-linear relationship between $\delta^{18}O_v$ and specific humidity is also depicted (Figure 3 d). We note for low temperatures and specific humidity values (below -20 °C and 0.1 g/kg) that the distribution of $\delta^{18}O_v$ values against temperature and specific humidity is curved towards higher $\delta^{18}O_v$ values, compared to this linear relationship. It is not certain if this deviation can be attributed to an

atmospheric process (such as a more important relative contribution of additional moisture sources at very low humidity compared to higher humidity levels), or if is due to a remaining observational bias or contamination that could not be identified despite our calibration and flagging processes.

| Season | | T (°C) | q (g/kg) | RH (%) | $\delta^{18}O_v$ (‰) | $\delta^{18}O_p$ (‰) | $\delta^2H_v$ (‰) | $\delta^2H_p$ (‰) | d-excess$_v$ (‰) | d-excess$_p$ (‰) |
|---|---|---|---|---|---|---|---|---|---|---|
| **Winter (DJF)** | **Mean** | **-29.3** | **0.32** | **77.5** | **-44.0** | **-30.0** | **-325.2** | **-237.9** | **27.2** | **1.9** |
| | **St. dev.** | 5.9 | 0.22 | 6.3 | 3.6 | 4.8 | 26.0 | 33.9 | 9.4 | 12.0 |
| | **Min.** | -40.9 | 0.06 | 60.6 | -52.6 | -41.3 | -378.2 | -324.6 | 2.6 | -39.7 |
| | **Max.** | -8.23 | 2.20 | 93.9 | -32.9 | -12.5 | -244.6 | -140.0 | 53.3 | 15.6 |
| | **N** | 684 | 620 | 724 | 550 | 43 | 550 | 43 | 550 | 43 |
| **Spring (MAM)** | **Mean** | **-14.7** | **1.48** | **86.0** | **-40.4** | **-24.4** | **-294.9** | **-195.7** | **28.1** | **-0.4** |
| | **St. dev.** | 9.4 | 0.98 | 6.2 | 5.3 | 5.3 | 35.7 | 35.5 | 9.8 | 19.5 |
| | **Min.** | -41.1 | 0.06 | 66.1 | -53.2 | -36.8 | -381.8 | -291.3 | 9.0 | -56.1 |
| | **Max.** | 3.5 | 4.63 | 101.2 | -27.0 | -8.7 | -203.9 | -108.3 | 62.1 | 33.1 |
| | **N** | 736 | 661 | 736 | 650 | 46 | 650 | 46 | 650 | 46 |
| **Summer (JJA)** | **Mean** | **6.9** | **5.56** | **83.7** | **-28.8** | **-15.9** | **-211.3** | **-122.2** | **19.1** | **4.6** |
| | **St. dev.** | 4.6 | 1.61 | 10.5 | 2.6 | 2.4 | 19.1 | 18.1 | 5.9 | 6.8 |
| | **Min.** | -4.9 | 2.41 | 40.3 | -36.7 | -21.7 | -274.1 | -166.0 | 6.7 | -12.2 |
| | **Max.** | 21.5 | 11.99 | 101.4 | -20.9 | -10.1 | -155.3 | -83.9 | 37.8 | 15.2 |
| | **N** | 748 | 583 | 748 | 577 | 61 | 577 | 61 | 577 | 61 |
| **Autumn (SON)** | **Mean** | **-8.6** | **2.41** | **88.3** | **-34.3** | **-19.1** | **-250.7** | **-147.4** | **23.9** | **5.7** |
| | **St. dev.** | 11.6 | 1.90 | 7.4 | 7.2 | 6.2 | 50.4 | 45.2 | 9.4 | 11.4 |
| | **Min.** | -37.4 | 0.12 | 61.0 | -53.1 | -31.1 | -370.5 | -252.8 | 4.3 | -60.9 |
| | **Max.** | 17.6 | 8.95 | 101.9 | -20.8 | -2.3 | -162.4 | -79.3 | 56.1 | 19.6 |
| | **N** | 728 | 618 | 726 | 618 | 59 | 618 | 59 | 618 | 59 |
| **All** | **Mean** | **-11.0** | **2.28** | **83.9** | **-36.9** | **-21.6** | **-270.3** | **-169.3** | **24.6** | **3.3** |
| | **St. dev.** | 15.4 | 2.33 | 8.8 | 7.6 | 7.1 | 55.1 | 55.6 | 9.5 | 13.0 |
| | **Min.** | -41.1 | 0.06 | 40.3 | -53.2 | -41.3 | -381.8 | -324.6 | 2.6 | -60.9 |
| | **Max.** | 21.5 | 11.99 | 101.9 | -20.8 | -2.3 | -155.3 | -79.3 | 62.1 | 33.1 |
| | **N** | 2896 | 2482 | 2934 | 2395 | 209 | 2395 | 209 | 2395 | 209 |

**Table 1: Averaged values, standard deviations, minima, maxima and number of values for all seasons and for the whole dataset within the period 2015-07-01 to 2017-06-30, with 6 hours averaged dataset, for air temperature (°C), specific humidity q (/kg), relative humidity (%), $\delta^{18}O_v$ (‰), $\delta^{18}O_P$ (‰), $\delta^2H_v$ (‰), $\delta^2H_p$ (‰) d-excess$_v$ (‰) and d-excess$_p$ (‰).**

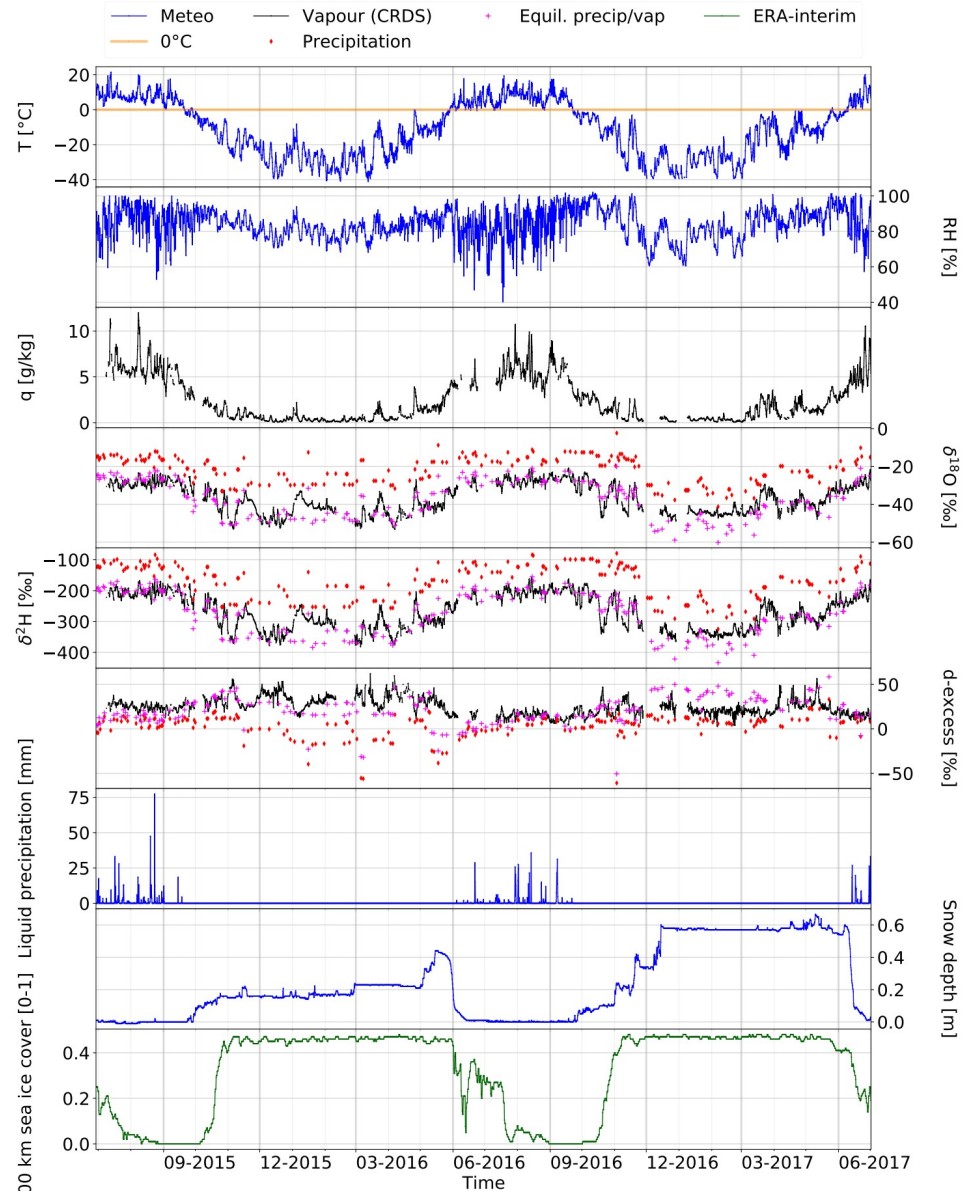

Figure 2: Time series of observations at Samoylov for the period 2015-07-01 to 2017-07-01 for air temperature (°C), relative humidity (%), specific humidity q (g kg$^{-1}$), **δ$^{18}$O** (‰), **δ$^2$H** (‰) and d-excess (‰), liquid precipitation amount (mm), depth of the snow cover (m) and fraction of the surrounding 500 km area covered by sea ice (from 0 to 1 without unit). Plain lines indicate the continuous parameters, displayed in blue for data recorded by the meteorological station, in black for data recorded by the water vapour analyser and in green for data from the ERA-interim reanalyses database. For the temperature, the plain horizontal orange line indicates the 0 °C value. Red diamonds (purple crosses) represent the discrete isotopic measurements from precipitation samples (respectively the

**resulting theoretical vapour isotopic composition considering equilibrium fractionation from precipitation). The temporal resolution is of 6 hours for all parameters, except for the precipitation and the resulting theoretical vapour isotopic composition (daily averages of event-based samples) and for the daily averaged sea ice cover.**

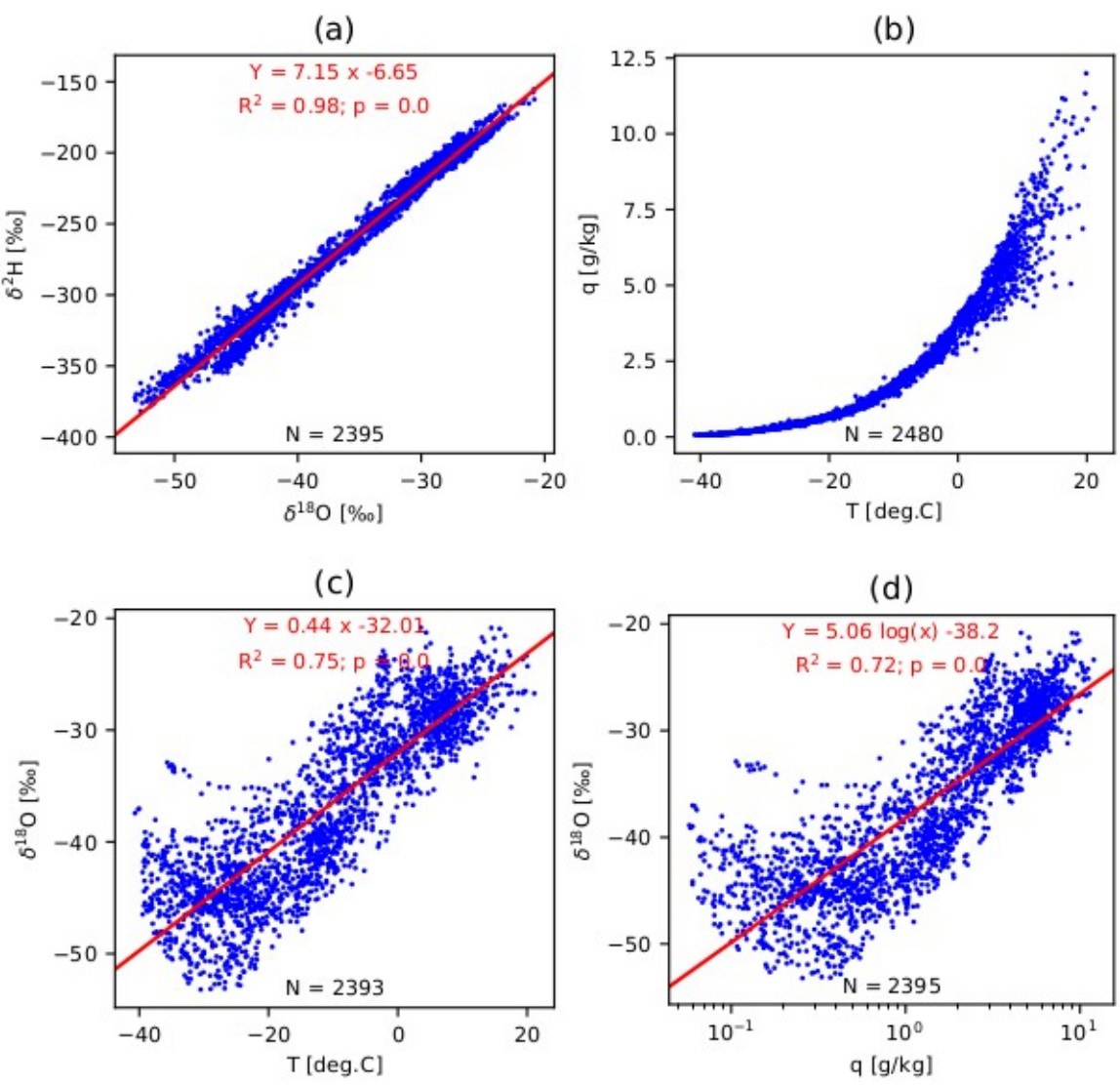

**Figure 3: Co-variations of water vapour isotopic composition and meteorological parameters, for the complete period 2015-07-01 to 2017-07-01. Red lines represent the best estimate of the linear regression.**

## 3.2 Seasonal cycle

A clear average seasonal cycle is observed (Figure 4 and Table 1) for temperature, specific humidity and relative humidity as
well as for the water vapour isotopic composition ($\delta_v$-values and d-excess$_v$). The $\delta^2H_v$ seasonal variations (not shown) are
very similar to the $\delta^{18}O_v$ seasonal variations (correlation coefficient $R^2$ =1.0, p<0.01, with a slope of 7.34 considering the
monthly averages). At the first order, the temperature depicts similar variations than the $\delta_v$-values, with a maximum during
summer compared to winter ($\delta^{18}O_v$/T slope of 0.45 ‰ °C$^{-1}$ considering the monthly averages, $R^2$ =0.92, p<0.01, Figure 4 and
Table 2).

A summer peak of temperature and specific humidity is reached in July (monthly averaged values of +8.1 +/- 4.2 °C and 6.1
+/- 1.4 g/kg). Significantly colder and drier specific humidity conditions are already measured in September (about +3.6 +/-
5.3 °C and 4.6 +/- 1.6 g/kg lower). For $\delta_v$-values however, a plateau of maximal values is extending from July to September,
with monthly mean $\delta^{18}O_v$ values between -28.0 +/- 2.4 and 28.5  +/- 2.2 ‰.

For the studied period, the lowest temperatures are measured in December, January and February (with respective monthly
means of -30.1 +/- 6.0 °C, -27.1 +/- 5.1 °C and -30.7 +/- 5.5 °C). Very dry air is observed during whole winter (specific
humidity below 1.0 g/kg from November to February. Two minima of specific humidity are measured in February and
December, with 0.26 +/- 0.2 and 0.30 +/- 0.2 g/kg respectively, corresponding to minimum values of monthly averaged
$\delta^{18}O_v$ (-47.1 ‰ +/- 2.5 and -44.1 +/- 2.3 ‰ for December and February, respectively). The difference in $\delta^{18}O_v$ between
December and February are hardly significant regarding the precision of our observations at very low humidity.

Transitions between the summer and winter regimes are observed for temperature, specific humidity and $\delta^{18}O_v$ in March to
June for spring and September to November for autumn. We note a temperature, humidity and $\delta^{18}O_v$ increase at the
beginning of March (+10.5 °C, +0.8 g/kg and +2.7 ‰ difference in monthly average between February and March), stagnant
until mid-April, before a sharp increase towards summer values until July. This increase of the monthly average values of
temperature, specific humidity and $\delta^{18}O_v$ in March is accompanied by a large temporal variability compared to the winter
325 months (Figure 4). This suggests that this early-spring transition is primarily linked with an increase of the synoptic
variability.

| Month | T (°C) | q (g/kg) | RH (%) | $\delta^{18}O_v$ (‰) | $\delta^2H_v$ (‰) | d-excess$_v$ (‰) |
|---|---|---|---|---|---|---|
| **January** | -27.1 +/- 5.1 | 0.41 +/- 0.,3 | 79.2 +/- 6.6 | -41.5 +/- 3.2 | -308.6 +/- 26.3 | 23.3 +/- 5.9 |
| **February** | -30.7 +/- 5.5 | 0.26 +/- 0.2 | 77.2 +/- 5.3 | -44.1 +/- 2.3 | -329.2 +/- 21.4 | 23.4 +/- 8.9 |
| **March** | -20.2 +/- 10.6 | 1.04 +/- 0.9 | 83.5 +/- 6.9 | -41.4 +/- 5.8 | -305.9 +/- 39.4 | 25.5 +/- 10.4 |
| **April** | -15.8 +/- 7.6 | 1.31 +/- 0.9 | 85.4 +/- 4.9 | -41.4 +/- 5.6 | -298.8 +/- 36.8 | 32.7 +/- 9.3 |
| **May** | -8.3 +/- 4.8 | 2.04 +/- 0.9 | 89.2 +/- 5.0 | -38.6 +/- 4.0 | -281.2 +/- 25.0 | 27.3 +/- 8.2 |
| **June** | 4.7 +/- 4.9 | 4.55 +/- 1.5 | 83.1 +/- 10.3 | -30.3 +/- 2.6 | -226.6 +/- 19.0 | 15.9 +/- 3.2 |
| **July** | 8.1 +/- 4.2 | 6.1 +/- 1.4 | 82.8 +/- 11.5 | -28.5 +/- 2.2 | -207.4 +/- 14.2 | 20.5 +/- 7.6 |
| **August** | 7.9 +/- 3.6 | 5.9 +/- 1.5 | 85.2 +/- 9.5 | -28.0 +/- 2.4 | -203.6 +/- 15.8 | 20.2 +/- 5.2 |
| **September** | 3.6 +/- 5.3 | 4.6 +/- 1.6 | 88.6 +/- 8.2 | -28.3 +/- 3.4 | -207.5 +/- 22.0 | 18.9 +/- 7.3 |
| **October** | -8.7 +/- 4.8 | 1.9 +/- 0.8 | 91.6 +/- 5.0 | -34.3 +/- 6.4 | -250.2 +/- 44.3 | 24.0 +/- 8.9 |
| **November** | -20.8 +/- 7.5 | 0.8 +/- 0.6 | 84.6 +/- 6.9 | -40.2 +/- 5.8 | -293.2 +/- 38.6 | 28.5 +/- 9.3 |
| **December** | -30.1 +/- 6.0 | 0.3 +/- 0.2 | 76.2 +/- 6.4 | -47.1 +/- 2.5 | -341.2 +/- 16.1 | 35.2 +/- 8.3 |

**Table 2: Monthly averaged values and standard deviations for the period 2015-07-01 to 2017-06-30, for air temperature (°C), specific humidity q (/kg), relative humidity (%), $\delta^{18}O_v$ (‰), $\delta^2H_v$ (‰) and d-excess$_v$ (‰).**

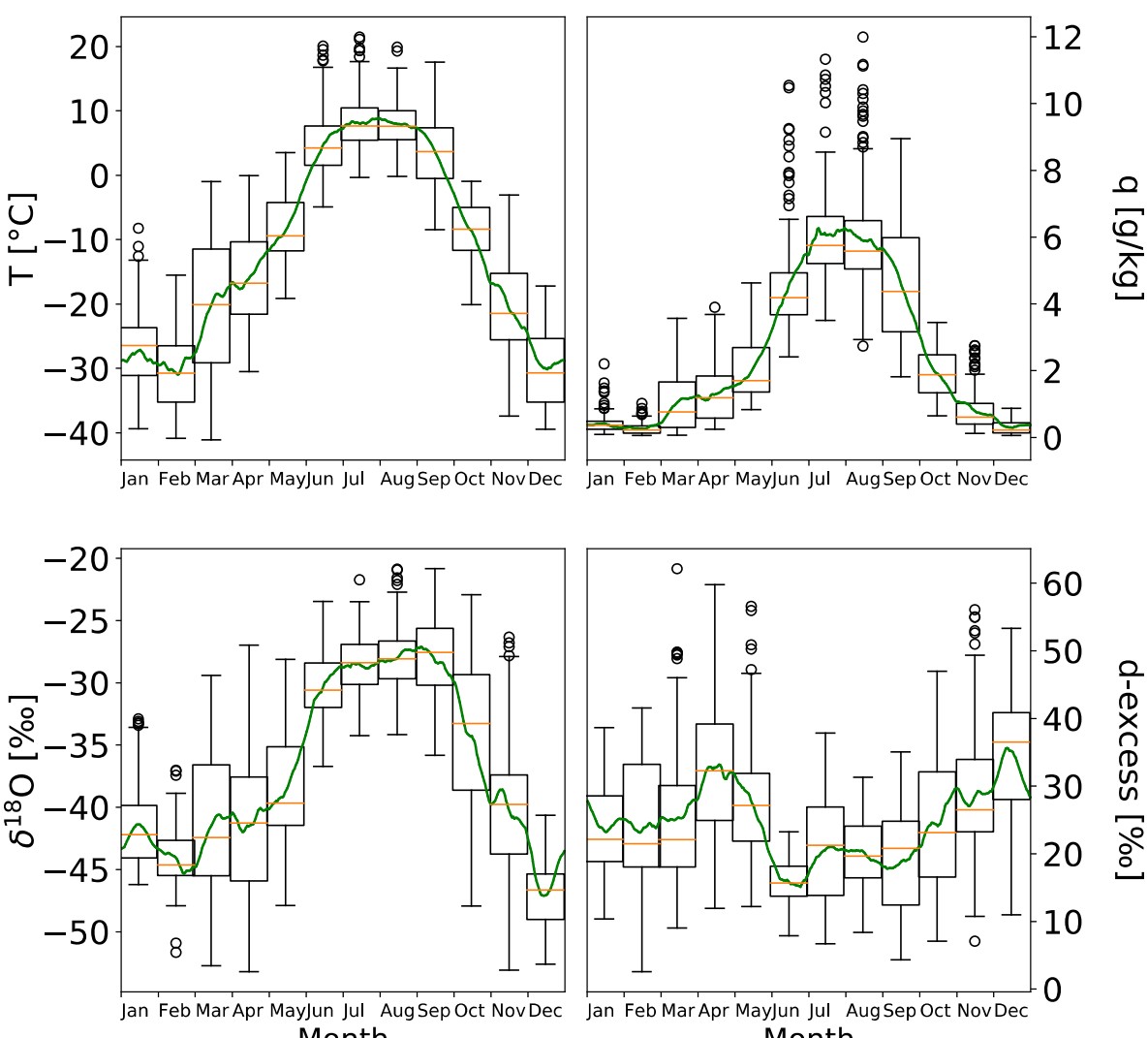

**Figure 4: Average seasonal cycle for the period 2015-07-01 to 2017-06-30, for (a) air temperature (°C), (b) specific humidity q (/kg), (c) δ¹⁸Oᵥ (‰) and (d) d-excessᵥ (‰). For box-plots, the boxes represent the first and third quartiles, and the orange bar represents the median; the whiskers represent the lowest (highest) datum still within 1.5 interquartile range (IQR) of the lower (upper) quartile and the outliers are represented as black circles.**

### 3.3 Synoptic variability

The synoptic variations, responsible for variations in the meteorological and water vapour isotopic signals at time scales from a few hours to a few weeks, are related to large-scale meteorological patterns and allow to investigate the influence of remote moisture sources. In order to distinguish these synoptic variations from potential diurnal variations, we use daily averaged values (Supplementary Figure 6). Compared to the seasonal variations, synoptic variations of specific humidity have a large amplitude in summer (on the order of 5 g kg$^{-1}$) but a low amplitude in winter (on the order of 1 to 2 g kg$^{-1}$). In contrast, the synoptic variations of air temperature have a large amplitude in winter (up to 20 °C) and a lower amplitude in summer (around 10 °C). This is coherent with the Clausius-Clapeyron relationship. Frequent episodes of strong relative humidity variations are observed from one day to the next between June and August (variations from down to 60 % to up to 100 %). During winter, the variations of relative humidity are slow (variations over several weeks) and of lower amplitude (changes in the order of magnitude of 20 %) than these fast summer variations.

### 3.4 Diurnal cycles

We expect diurnal variations of the local evaporation flux to cause diurnal variations of the water vapour isotopic composition. To evaluate the diurnal cycles of meteorological and water isotopic composition, we use datasets averaged at 1-hour temporal resolution. Some periods of relatively stable synoptic situation (with low horizontal wind speed) during spring and summer clearly show significant variations of temperature and relative humidity on the diurnal time scale. We compute the average diurnal cycle of temperature, specific and relative humidity and vapour isotopic composition for these stable synoptic situation periods over different seasons. As the synoptic activity can lead to important changes within a few hours, potentially hiding the signal associated to diurnal variations, a filtering method has been implemented to compute these average diurnal variations. We select periods for which the horizontal wind speed is always below 5 m s$^{-1}$ during at least 24 hours. We suppose that this criterion allows getting rid of the influence of important synoptic events changes and the variations of the observed ambient air are affected by local processes rather than large scale transport changes.

No diurnal cycle is seen in any of the parameters for winter and autumn (not presented), which is the period dominated by the polar night. In summer (see Supplementary Figure 5), an average temperature diurnal cycle is found for stable synoptic situation periods with an amplitude of respectively 3 °C, which is lower than the typical amplitude of synoptic variations during this season (around 10 °C, see Figure 2 and averaged diurnal cycles on Supplementary Figure 5). For these periods of stable synoptic condition, the average diurnal variations of specific humidity is not significant in summer compared to the synoptic variability (the amplitude of the mean cycle is lower than the standard deviation).

The observed summer average diurnal cycle of relative humidity observed (average amplitude of 13.4 % average amplitude) is probably due to the diurnal variations of temperature which are not followed by any diurnal variations of the specific humidity. As for the specific humidity signal, there are no significant average diurnal variations of any vapour isotopic signals ($\delta^{18}O_v$, $\delta^2H_v$ or d-excess$_v$) for neither season.

During spring, strong changes occur regarding temperature, specific humidity, isotopic composition or insolation within a few weeks. Analyses of the average diurnal cycle for days of stable synoptic situation over a three months period from

370 March to May do not reveal any significant diurnal variations of the specific humidity and vapour isotopic composition. However, a comparable analysis over the month of May, only, when the daily variations of the insolation is maximal, reveals significant average diurnal variations (Figure 5) for temperature (with an amplitude of 4.9 °C), relative humidity (amplitude of 5.4 %), specific humidity (amplitude of 0.4 g/kg), $\delta^{18}O_v$ (amplitude of 1.1 ‰) and $\delta^2H_v$ (amplitude of 10 ‰). The diurnal cycle is not significant for d-excess$_v$, as compared to the observed variability and also smaller than the instrumental

precision.

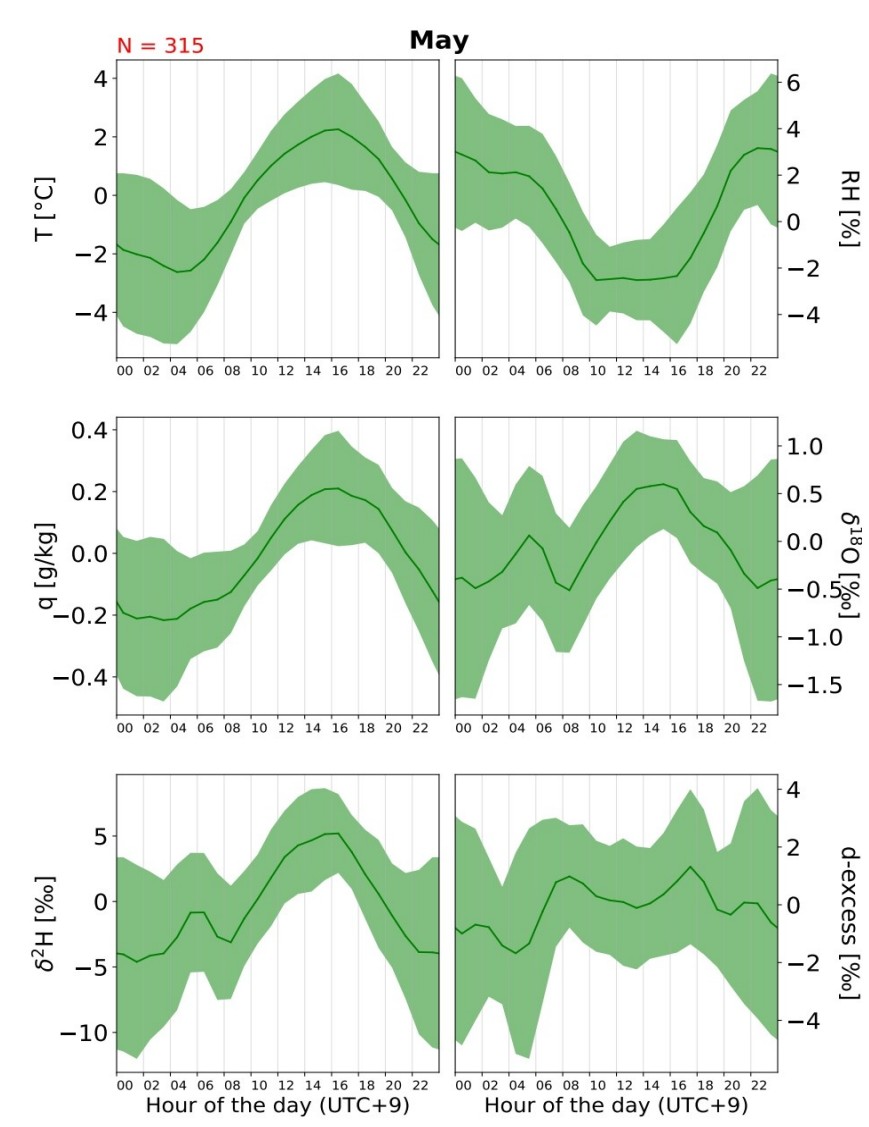

**Figure 5**: Average daily cycle in May, for the selected days of stable synoptic conditions (wind speed below 5 m.s[-1] over 24 hours), for T (°C), RH (%), q (g/kg), $\delta^{18}O_v$ (‰), $\delta^2H_v$ (‰) and d-excess$_v$ (‰). The daily cycles are calculated with the stacked anomalies compared to the daily average. The local time (UTC+9) is used to present the results.

## 4 Discussion

### 4.1 Temporal variations of the water isotopic composition

Comparing the precipitation and vapour isotopic composition reveals a large offset between the δ-values, as well as the d-excess values. To investigate the reason of this offset, we calculate a theoretical vapour isotopic composition from the precipitation samples. Important assumptions are made for this calculation, leading to uncertainties on the calculated theoretical isotopic values. : wWe consider use the temperature measured at 2 m height at the time of the sampling for the determination of the equilibrium fractionation coefficients. Considering a 24 hours uncertainty on the real time of the precipitation and consequently on the temperature leads to mean variations of 0.45 ‰, 4 ‰ and 0.7 ‰ in $\delta^{18}O$, $\delta^{2}H$ and d-excess, respectively. The temperature at which the precipitation is formed is also different from the temperature at 2 m. As the Arctic regions are mostly dominated by low clouds (Cesana et al., 2012), considering a wet adiabatic lapse rate of -6,5°C/ km and clouds at 2 km above the ground level, we estimate the uncertainty linked to the cloud elevation of 0.5 ‰, 4 ‰, and 0.6 ‰ in $\delta^{18}O$, $\delta^{2}H$ and d-excess, respectively. Finally, the main uncertainty is associated with the choice of an equilibrium fractionation between vapour and the condensed phase which can be either (liquid or solid. We chose to use fractionation coefficients we the liquid phase for positive temperatures and with the solid phase or for negative temperatures) using the temperature measured at 2 m height at the time of the sampling. Considering an uncertainty on this choice of coefficients for data with temperatures ranging from -10°C to +10°C would lead to mean differences for these data only of 3.8 ‰, 18 ‰ and 12 ‰ in $\delta^{18}O$, $\delta^{2}H$ and d-excess, respectively. Altogether, considering these uncertainties, Tthe calculated theoretical vapour $\delta^{18}O$ and $\delta^{2}H$ ($\delta^{18}O_{v,th}$ and $\delta^{2}H_{v,th}$) agree with the measured $\delta^{18}O_v$ and $\delta^{2}H_v$ during summer, spring and autumn (Figure 2). The calculated theoretical and measured summer and autumn d-excess values also match, showing that precipitation is in equilibrium with vapour during these seasons (Figure 2). However, large discrepancies remain between the winter and spring calculated theoretical and measured vapour d-excess signals (Figure 2), which can not only be explained by the uncertainty of the theoretical calculations, indicating either that the precipitation formed from a different moisture source than the measured vapour at the surface, and/or that kinetic fractionation occurs during the formation of precipitation during this period, as expected from the theory of snow crystal formation under supersaturation (Jouzel and Merlivat, 1984) (Jouzel and Merlivat, 1984).

The absence of a significant diurnal cycle in specific humidity and water vapour isotopic composition at our site in summer differs from observations at the more southerly located western Siberia station Kourovka (Bastrikov et al., 2014)(Bastrikov et al., 2014), where the summer signal is dominated by the diurnal cycle, which has been attributed to a strong contribution of the local evapotranspiration to the moisture budget. The absence of a clear diurnal cycle at the Samoylov site also differs from other high latitude polar locations, like the interior of Greenland and Antarctica, where strong diurnal cycles have been

recorded during summer (Casado et al., 2016; Ritter et al., 2016; Steen-Larsen et al., 2014)(Casado et al., 2016; Ritter et al., 2016; Steen-Larsen et al., 2014) and attributed to exchange processes between the vapour and the snow-covered surface, which is not the case for our site, where the surrounding area consists of vegetated areas, small lakes and the rivers network of the Lena delta in summer. The diurnal temperature variations at Samoylov are also small compared to observations at Kourovka.Since eddy covariance observations performed on the Samoylov Island (Helbig et al., 2013)(Helbig et al., 2013) have revealed maxima of evapotranspiration concomitant with peaks of net radiation in June, diurnal exchanges of moisture between the atmosphere and the surface exist in summer.  It is therefore unlikely that the absence of a diurnal cycle of specific humidity and vapour isotopic values at Samoylov could be explained by insufficiency diurnal variations of incoming radiations in summer. Another explanation would rather be that the isotopic signal of the evaporation flux is too similar to the boundary layer water vapour isotopic signal to cause a significant diurnal isotopic variation even in case of significant evaporation. Altogether, the diurnal cycles of the boundary layer specific humidity and its isotopic composition, which are too small to be identified among the variabilities linked with the synoptic activity, are not the appropriate parameter to evaluate the impact of local evaporation at our site.

Contrary to the rest of the year, a diurnal cycle of the specific humidity and water vapour isotopic composition is observed in May. This diurnal cycle might be caused by local moisture exchanges with the surface, as it is the case over the interior of Greenland and Antarctica in summer. In these polar locations, summer diurnal cycles of the water vapour isotopic composition has been related to sublimation of the snow cover when the insolation is high and condensation during the night. (Casado et al., 2016; Ritter et al., 2016; Steen-Larsen et al., 2014)(Casado et al., 2016; Ritter et al., 2016; Steen-Larsen et al., 2014).At our site, sublimation of the snow cover would be possible only until end of May, when the snow cover, composed of winter precipitation with $\delta^{18}O_v$ values situated between -20 and -30 ‰ is present. Its sublimation would create moisture with $\delta^{18}O_v$ values higher than the boundary layer water vapour (mean monthly $\delta^{18}O_v$ of -38.6 +/- 4.0 ‰), which is coherent with the observed diurnal variations. Such sublimation process is probably not sufficient to drive significant diurnal variations of the isotopic composition earlier in the season as the insolation is still low (and event less in winter during the polar night).

Concerning the seasonal variations of the water vapour isotopic composition, the $\delta^{18}O_v$ values are primarily driven by the variations of temperature and humidity, following the principles of Rayleigh distillation. A possible explanation of the enriched values of $\delta^{18}O_v$ observed at temperatures below -20°C is that the contribution of local moisture sources is strongly impacting on $\delta^{18}O_v$ signals in an already very dry air.

The late-autumn/early winter maximum of d-excess$_v$ observed at Samoylov is similar to water isotopic measurements from precipitation samples along the Lena river in Zhigansk (66.8°N, 123.4°E, 92 m above sea level, approximately 635 km south from Samoylov station) depicting a maximum of d-excess$_v$ from October to December (Kurita, 2011)(Kurita, 2011). This peak of d-excess$_v$ has been interpreted as a signal linked with the sea-ice growth in the Arctic Ocean and fast oceanic evaporation occurring at low relative humidity during this period. Maxima of d-excess$_v$ in late-autumn have also been observed in different locations, in particular around the North-Atlantic sector (Bonne et al., 2014; Steen-Larsen et al., 2015)

(Bonne et al., 2014; Steen-Larsen et al., 2015). The very low relative humidity values above the ocean where the evaporation takes place in this season could explain these high d-excess$_v$ values (Pfahl and Sodemann, 2014; Steen-Larsen et al., 2014) (Pfahl and Sodemann, 2014; Steen-Larsen et al., 2014). The other spike of d-excess$_v$ observed in April/May is not concomitant with low relative humidity. It might however be an indicator of sublimation processes taking place. During this period, temperatures are still low but rapidly rising (monthly average temperatures of -8.4 °C in May), snow cover is decreasing and the solar radiation increases with important diurnal variations. The interpretation of this high d-excess$_v$ period as an effect of sublimation agrees with our explanation of the diurnal cycle observed in May.

## 4.2 Moisture sources variations at the seasonal scale

Changes of the moisture sources can affect water stable isotopic composition. Firstly, due to the link between temperature difference and spatial transport: the longer the moisture has been transported from the evaporation source, the more distillation an air mass might have undergone on the way. Secondly, the meteorological conditions at which the evaporation takes place will influence the isotopic composition of the initial vapour. Finally, different (primary and secondary) evaporation sources can have different isotopic signals. We focus here on the seasonal changes of the moisture sources from the averaged outputs of the semi-Lagrangian backward trajectories simulation and moisture source diagnostics over different seasons (Figures 5 and 6).

The overall amount of evaporation is strongly season-dependent (Figure 6), which reflects in the seasonality of the specific humidity measured at our site. During the winter months (December, January, February; DJF), the area with the highest moisture uptakes is situated above the Barents and Norwegian Seas (Figure 6 a). The moisture origin is therefore particularly distant from our site and long-distance transport dominates over local processes. For spring (March, April, May; MAM) and autumn (September, October, November; SON), an enhanced continental evaporation is observed over a large part of Siberia, contrary to winter. The main moisture uptake is located along the northern part of the Lena river basin. Minor oceanic sources are also revealed in some areas of the Arctic Ocean (Figure 6 d). In spring, only parts of the Barents and Norwegian Seas depict moisture uptake, while it is also the case for the Laptev, Kara and East Siberian Seas in autumn (Figure 6 b), which is coherent with the sea ice covering these last regions in spring, preventing oceanic evaporation, but not during the first autumn months. In summer (June, July, August; JJA), more air masses originate from the western Arctic Ocean than in winter. Despite the large ice-free surface in the Arctic during summer, the moisture uptake remains very low over the Arctic Ocean compared to the evapotranspiration taking place locally over the continent (Figure 6 c). The most predominant moisture uptake occurs locally above continental areas, mainly south-west of our site (Figure 6 c). Local vegetation, lakes, ponds and rivers are the potential sources of evapotranspiration which can generate such regional moisture uptake in summer (due to the absence of snow and ice covers in this season).

For all seasons, the average difference between evaporation and precipitation shows a general pattern of dominant precipitation at high latitudes and dominant evaporation at low latitudes (Figure 7). The limit at which the precipitation

prevails over evaporation is different for continental and oceanic regions. It is relatively stable around 50°N above the ocean, but its latitude varies from 60°N in summer and 45°N in winter over the continent (Figure 7). At our site, precipitation is always predominant over evaporation at any season.

The local moisture uptake in the region surrounding the station in particular in summer can, however, contribute to the water vapour isotopic signal, even if evaporation is lower than precipitation. Local evapotranspiration sources exist in summer,
such as the vegetation, ponds and lakes and the Lena river. As these sources are not active during the cold seasons, sublimation of the snow cover can act as a local moisture source, as previously suggested from the diurnal cycle revealed in spring.

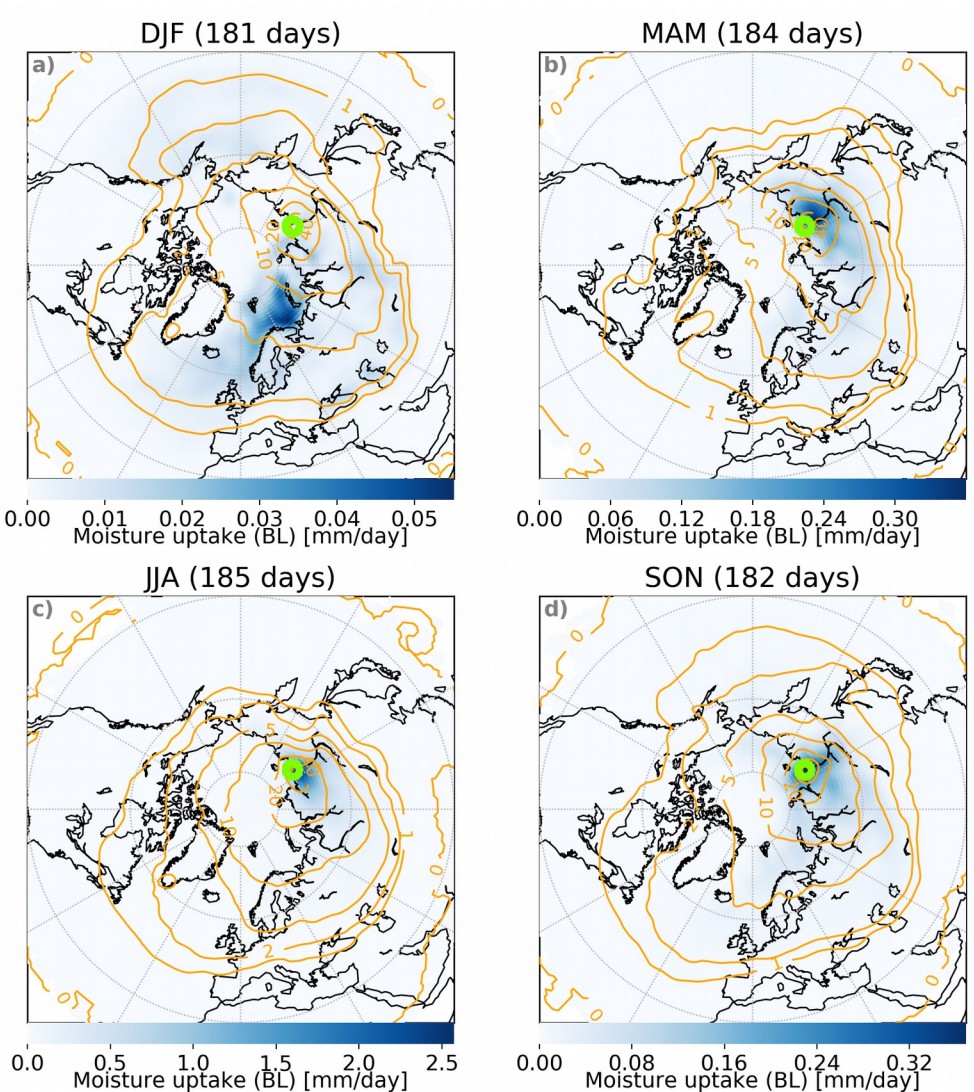

**Figure 6: Seasonal averages of boundary layer moisture uptake in mm/day for the period 2015-07-01 to 2017-07-01**
**(North pole Lambert azimuthal projection). For (a) winter, (b) spring, (c) summer and (d) autumn periods. The**

orange contour lines indicate the trajectories locations (as percentage of maximum value: 80, 40, 20, 10, 5, 2, 1 and 0 %). The green circles indicate the location of the Samoylov station.

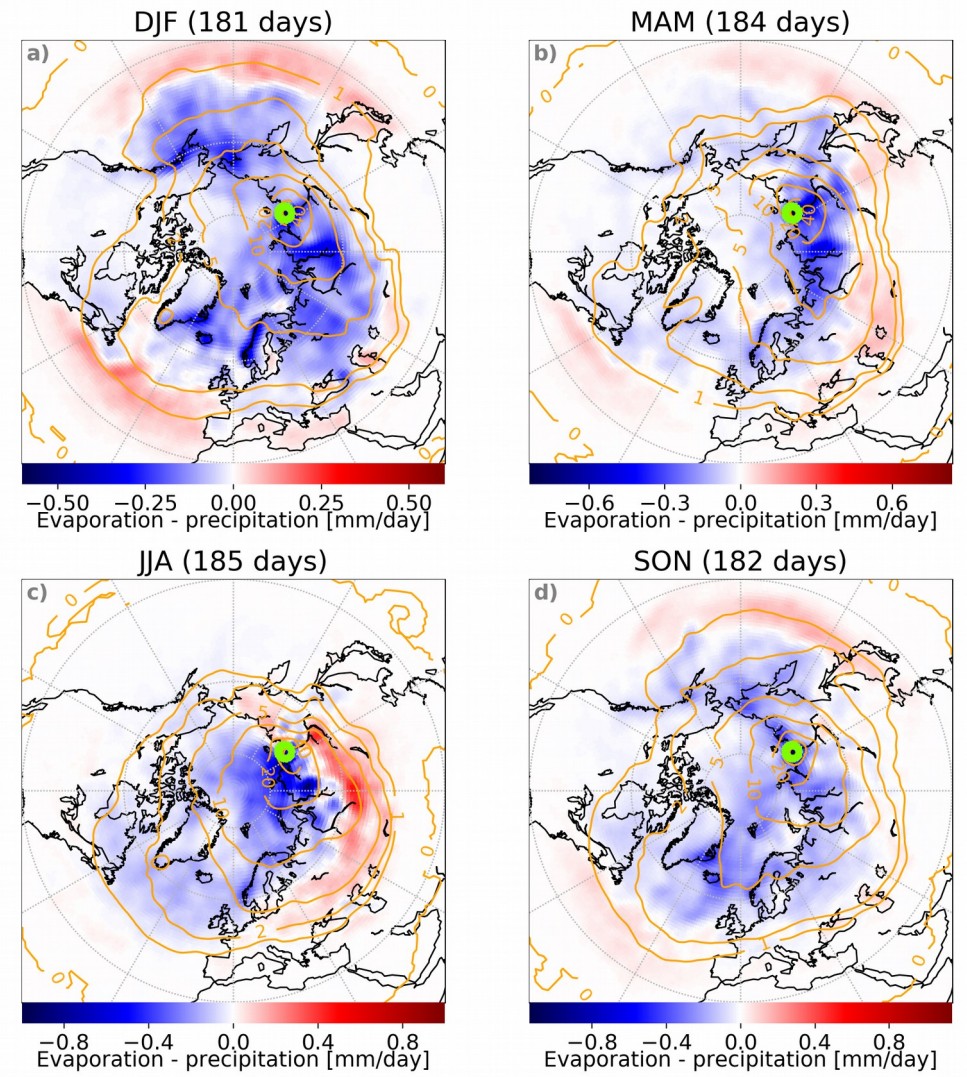

Figure 7: Same as Figure 6, with colours indicating the seasonal averages of evaporation minus precipitation (in mm/day): red (blue) colours indicate regions where evaporation is stronger (respectively weaker) than precipitation.

### 4.3 Seasonal versus synoptic variabilities and water vapour isotopic composition

### 4.3.1 Influence of the wind origin

To evaluate the impact of these seasonal moisture origin changes on the water vapour isotopic composition, we focus in this section on the statistical distribution of specific humidity and isotopes as a function of the wind direction observed at our site (Figure 8).

The summer season (JJA) presents a relatively homogeneous wind distribution: winds from all sectors represent between 4.3 and 8.6 % of the observations (Figure 8). No wind sector is exclusively associated to a single range of specific humidity or 505 water vapour isotopic values, but the frequencies of occurrences for these values still vary with the wind direction. However, the highest values of specific humidity originate from the south-south-east, thus along the Lena river basin (Figure 8 a). The most enriched $\delta^{18}O_v$ values and the lowest d-excess$_v$ values derive from a wide range of western sectors, while a higher proportion of depleted $\delta^{18}O_v$ values and the highest d-excess$_v$ values are associated with air masses originating from the east (Figure 8 b,c). This provides a potential way to differentiate between the moisture sources originating from the Atlantic and 510 the Pacific sectors.

We investigate the coldest months, from December to April (DJFAM), as they are the relevant months to contribute to the interpretation of the paleoclimate data retrieved from ice wedges in our research area (the ice wedges being formed from the melting of the snow deposited during this period). Contrary to the summer period, the cold months wind distribution exhibits a strongly predominant situation, with most winds originating from the south-south-east (20.6 %) and the south sectors (16.3 515 %) i.e. along the orographic barrier of the Verkhoyansk mountains. As for the summer season, all wind sectors present similar ranges of specific humidity and water vapour isotopic composition, but with different distributions. The south and south-south-east sectors are associated with a stronger proportion of very dry air (between 0.1 and 0.6 g/kg, Figure 8 d), and of the isotopically most depleted vapour ($\delta^{18}O_v$ below -45.7 ‰, Figure 7 e), compared to the other sectors. These very low $\delta^{18}O_v$ values associated with air masses originating from the continent are consistent with the absence of moisture uptakes 520 above a large part of the continent during this period (Figure 7). These air masses therefore undergo a strong isotopic distillation above the continent before reaching our site. For the air masses originating from the south-west, north-west or north-east, some significantly closer moisture sources can contribute to the isotopic composition of air masses (significant moisture uptakes exist in the north Atlantic and North Pacific sectors, as depicted on Figure 7). This is reflected in a higher proportion of high $\delta^{18}O_v$ values for these wind sectors, compared to the air masses originating from the south and south-525 south-east sectors. The most enriched air masses ($\delta^{18}O_v$ above -30.7 ‰) originate from the North-West sector (Figure 8 e).

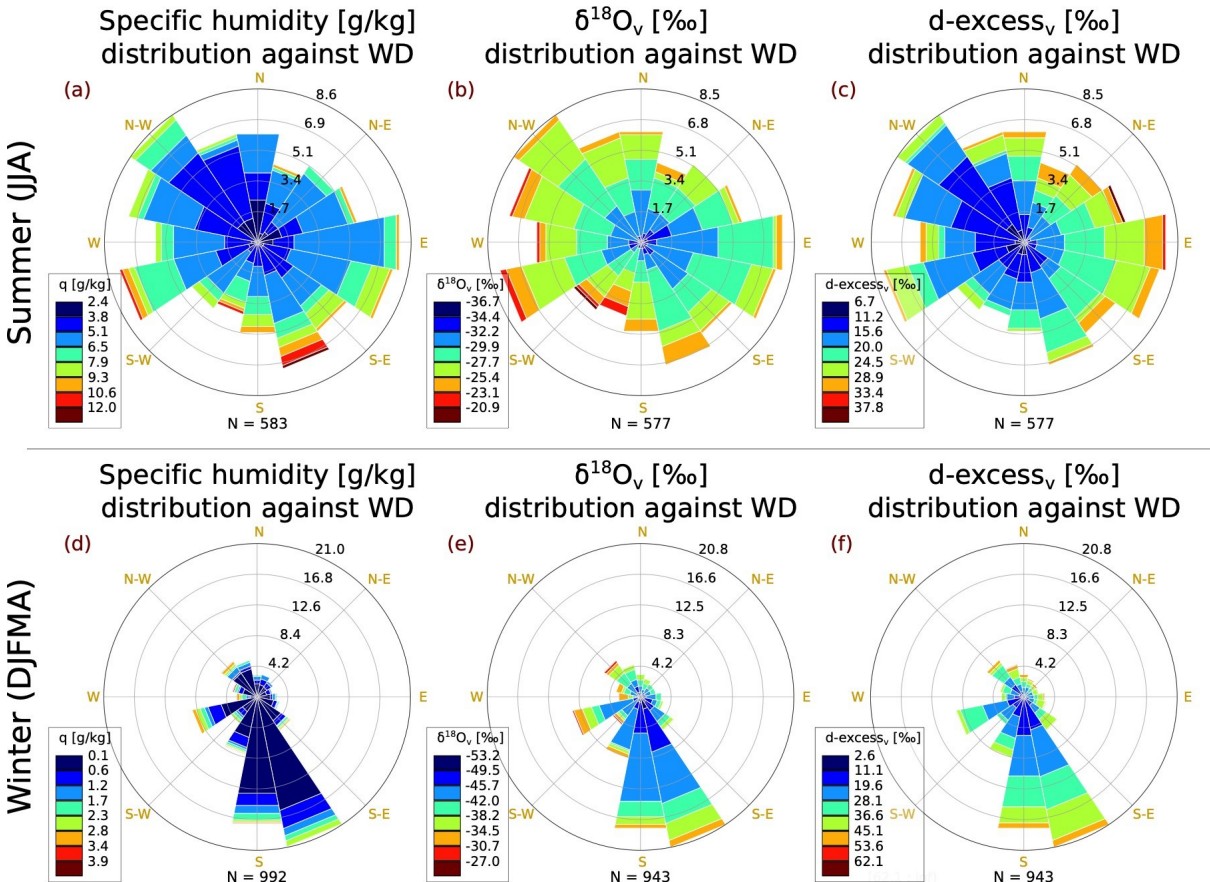

**Figure 8: Percent distribution of (a,d) specific humidity, (b,e) $\delta^{18}O_v$ and (c,f) d-excess$_v$: for (a, b, c) the summer months only (June to August) and (d, e, f) the cold months only (from December to April).The percent distributions are given with respect to the different wind directions (WD).**

**4.3.2 Influence of the moisture source**

As previously described, notable differences of water vapour isotopic composition exist between wind sectors during the winter season, in particular for the extreme high and low $\delta^{18}O_v$ values. As the local wind direction only provides information on the final step of air masses transport, we further investigate the outputs of the moisture sources diagnostics associated with synoptic events of extreme isotopic values.

For the coldest months (December to April), a selection of extreme values based on the distance to a 60-days running average for temperature, the logarithm of specific humidity, $\delta^{18}O_v$ and d-excess$_v$ shows strong similarities for the different parameters considered (Figure 9 and Supplementary Tables 1 and 2). Many synoptic events of short duration occurring on a daily time-scale, are selected independently for the different parameters. However, as seen on Figure 9, some patterns emerge with periods of several weeks where extrema are identified on multiple parameters, even if the extrema are not

always happening at the exact same date for all parameters (see Tables 1 and 2). Two typical situations are predominant: on one side, a typical pattern of high temperature, specific humidity and $\delta^{18}O_v$ but low d-excess$_v$; on the other side, an opposite pattern characterised by low temperature, specific humidity and $\delta^{18}O_v$ but high d-excess$_v$. Since the long-lasting events dominate this selection and are usually selected for all parameters, the associated average moisture source diagnostics are very similar for all parameters (Figure 10).

The first pattern (Figure 10 a,c,e,g), characterised by low temperature, specific humidity and $\delta^{18}O_v$ but high d-excess$_v$, is associated with local air masses, predominantly originating from the east. Precipitation is stronger than evaporation over whole Northern Eurasia as well as all polar and sub-polar Oceans. Evaporation is only dominant in very remote locations, like the subtropical North Pacific Ocean and the mid-latitudinal Atlantic Ocean.

The second transport pattern (Figure 10 b,d,f,h), characterized by high temperature, specific humidity and $\delta^{18}O_v$ but low d-excess$_v$, is associated with long-range transport of air masses originating from the west. Air masses originate from North-western Eurasia, up to the Lena river on the eastern limit, and from the southwestern Arctic Ocean. Precipitation is stronger than evaporation above the northernmost sectors of western Eurasia and over a large part of the Arctic Ocean. However, evaporation is stronger than precipitation over the northern European seas (North, Norwegian, Barents and Baltic Seas) and for continental areas located up to 60°N. The fast transport of air masses with only moderate precipitation brings moisture from the sources to our site with relatively high $\delta^{18}O_v$ values for this season.

The moisture sources are much more distant for the first (low temperature) pattern than for the second (high temperature) pattern. With a large temperature difference between the source of moisture and our observation site, a strong isotopic distillation can take place, which would explain that the $\delta^{18}O_v$ values associated to these atmospheric transport patterns are lower for the first pattern (with low temperature and distant sources).

The second pattern of high temperature in winter previously described presents some similarities with the average summer situation, such as a predominance of air masses originating from the west. If the limit of net evaporation compared to precipitation is situated at higher latitudes for the high temperatures pattern than for the low temperatures pattern, there are still fewer net sources of evaporation over the continent than in summer.

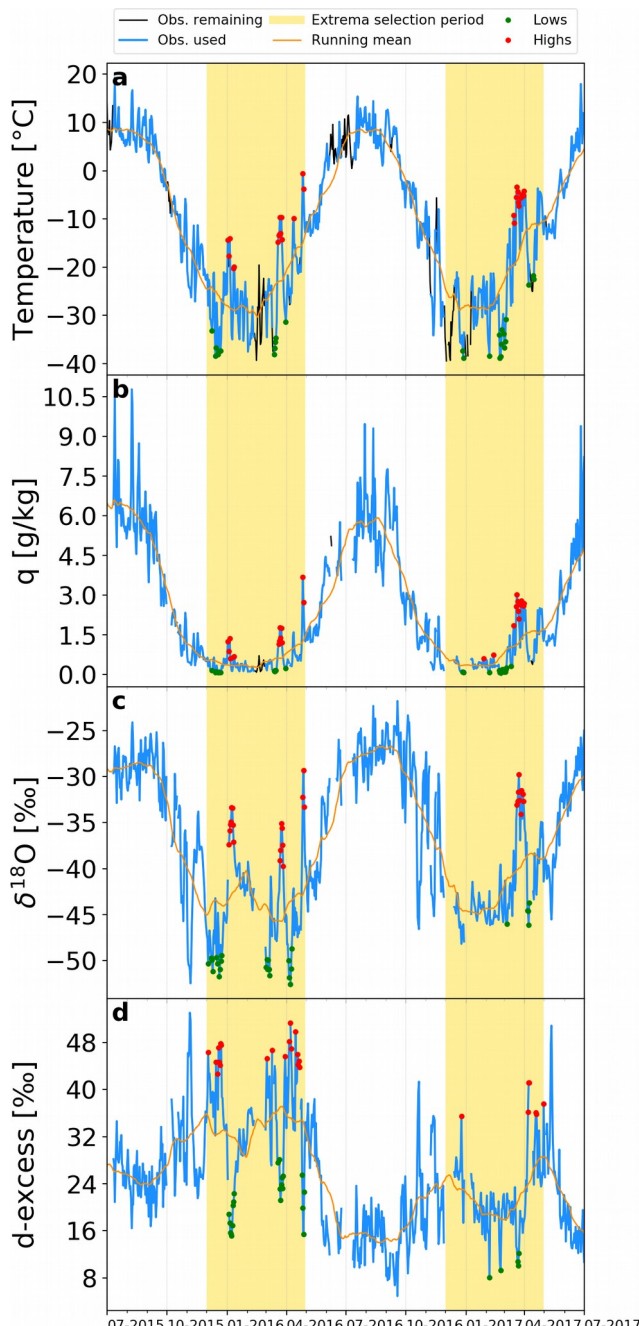

**Figure 9: Time series of (a) temperature, (b) specific humidity, (c) $\delta^{18}O_v$ and (d) d-excess$_v$. The daily averaged data are depicted in blue for the data used to compute the 60-days running average (orange line) and in black for the remaining data (not used in the running average calculation). The yellow shade represents the period over which extrema (depicted as red and green dots for high and low extrema, respectively) are considered: from December to April for each year.**

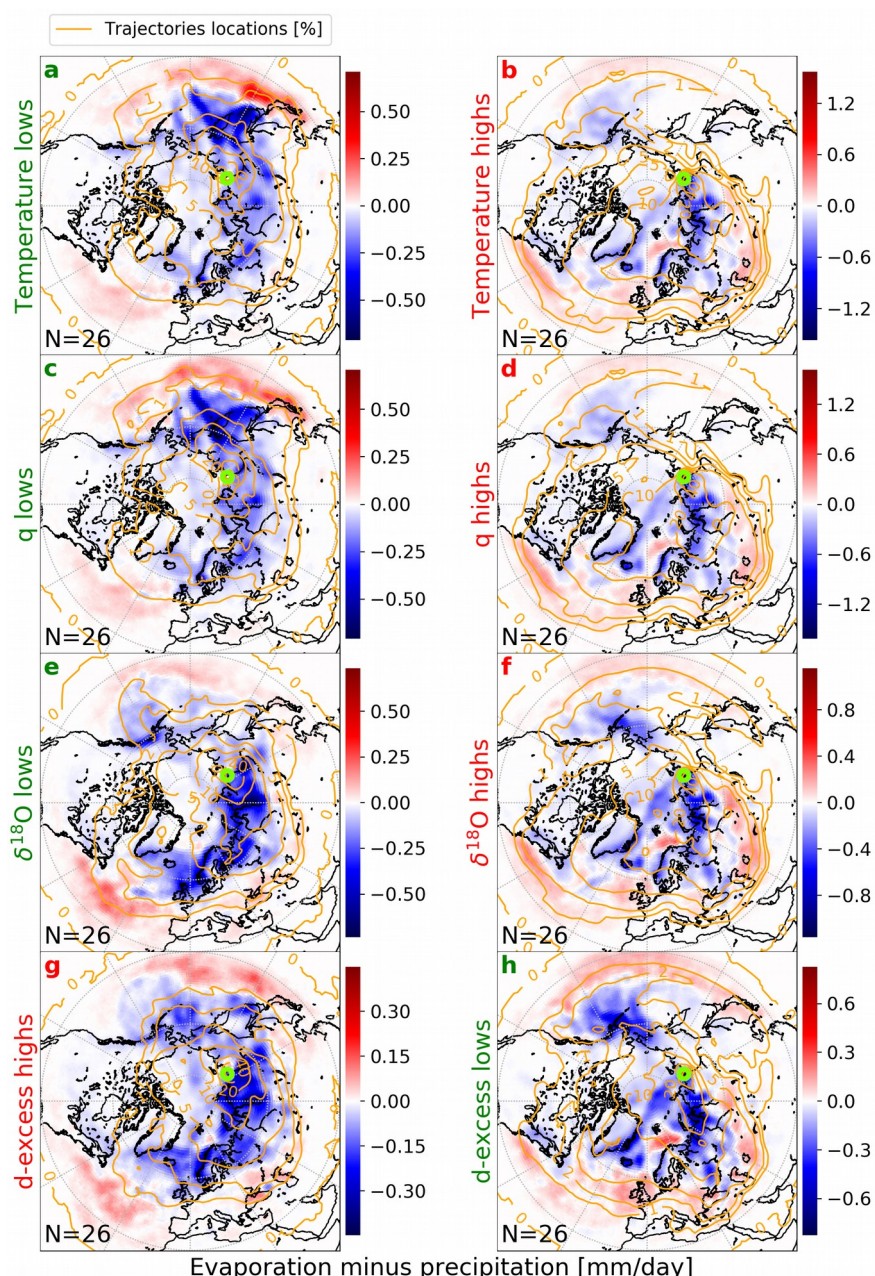

**Figure 10: Average "evaporation minus precipitation" (North pole Lambert azimuthal projection) for the selection of low and high extrema over the December to April period, as depicted on Figure 9: for temperature lows (a) and highs (b), for specific humidity lows (c) and highs (d), for $\delta^{18}O_v$ lows (e) and highs (f) and for d-excess$_v$ highs (g) and lows (h). The orange contour lines indicate the trajectories locations (as percentage of maximum value: 80, 40, 20, 10, 5, 2, 1 and 0 %). The green circles indicate the location of the Samoylov station.**

## 5 Conclusions

This study presents two years of *in situ* continuous water vapour isotopic observations in the Siberian Arctic on the Samoylov Island in the Lena river delta, starting in July 2015.

This new dataset provides information about the moisture isotopic composition which are complementary to the precipitation samples isotopic composition, as they also provide information for dry periods when no precipitation falls. It also allows comparing the water isotopic composition in both phases and distinguish periods when precipitation and vapour are at

585 equilibrium (in summer and autumn) or out of equilibrium (during winter and spring). This comparison therefore highlights the kinetic fractionation processes occurring during the formation of snow in the cold periods.

The water vapour isotope dynamics is dominated by seasonal and synoptic variations. During the coldest months, the observed humidity and water vapour isotopic composition are comparable to summer observations on the East Antarctic plateau. The diversity of isotopic signals associated with long-range transported moisture from various remote origins has a

590 strong imprint on the vapour isotopic composition observed at the Samoylov station.

In summer, the observed diurnal cycle of temperature has a low amplitude and is not clearly mirrored by diurnal cycles of the specific humidity and vapour isotopic compositions, but overwhelmed by variations linked to synoptic activity. Our data indicate either that the local sources of humidity are not strongly varying at a diurnal scale or that the isotopic signal of a local source is not distinguishable from the remotely transported moisture in this season. The situation is different during

spring, when the insolation is strongly varying between day and night and the surface is still covered by snow. During this period, significant diurnal variations of the specific humidity and water vapour isotopic composition are revealed, which indicates moisture exchange processes between the boundary layer atmosphere and the surface, with a potential influence of snow cover sublimation. It is however difficult to independently evaluate the imprints on the vapour isotopic composition of each potential individual local moisture source (vegetated or snow-covered land, open or frozen river, open over sea ice

covered Laptev sea), as many changes of the surface cover happen simultaneously within a few weeks at this period, as well as in autumn, and may affect the local hydrological cycle.

Evaluations of the moisture sources, based on semi-Lagrangian back-trajectories simulations, show small but significant changes of air masses origin at the seasonal scale. Depending on the seasons, areas of air masses origins differently contribute to the moisture balance of our study area. In the region surrounding Samoylov station, significant moisture uptake

from the surface is found in summer and to a lower extent in spring and autumn, which contributes to the observed vapour isotopic signal, while such local surface moisture uptake is almost inexistent in winter. Despite the local surface moisture uptake, the precipitation predominates over surface moisture uptakes on average for all seasons over the region surrounding our site. Surface moisture uptakes are predominant over precipitation for some remote locations only, over the Atlantic and Pacific Oceans as well as over the Eurasian continent, south of a limit which moves southwards during the cold seasons and

northward during the warm seasons. The main moisture sources are therefore generally more distant from our research area in winter than in summer.

On the synoptic time scale, it is not possible to exclusively attribute all observed vapour isotopic signals to a specific moisture source. However, there are statistical differences in the vapour isotopic distribution related with the moisture origin which provides a potential way to identify moisture source changes from recorded isotopic compositions. In summer, the most enriched $\delta^{18}O_v$ values and the lowest d-excess$_v$ values are associated with air masses originating from the west (Atlantic sector), while a higher proportion of depleted $\delta^{18}O_v$ values and the highest d-excess$_v$ values are associated with air masses originating from the east (Pacific sector). During the cold months (December to April), cold and dry air with isotopically depleted vapour and high d-excess$_v$ is associated with stagnant air masses above northern Eurasia, and vapour originating either from the Pacific (the most common situation) or from the Atlantic sectors. The significant contribution of eastbound moisture source from the Pacific regions is a new finding of this study. During the same period, an opposite situation of warm, moist air with isotopically enriched vapour and low values of d-excess$_v$ is typically associated with air masses originating from the North-Atlantic basin and transported via westerlies rapidly towards our site. Such events are frequently observed between March and mid-April, associated with an early spring increase of temperature, humidity and $\delta^{18}O_v$.

Our study contributes to an improved knowledge of the variations of water isotopic composition from the seasonal to the synoptic scale in the eastern Arctic continental region. It will help improving the interpretation of water isotopes paleoclimate proxies based on ice wedges retrieved in the region (e.g., Meyer et al., 2015)~~(e.g., Meyer et al., 2015)~~. It contributes to the understanding of the moisture sources and atmospheric transport processes. It is also a baseline for studies of future changes in the region, as many hydrological changes are expected in the region, in particular with retreating sea ice over the Arctic Ocean, which might trigger more effective contributions from this moisture source and affect the water isotopic composition in vapour and precipitation.

**Appendices**

Supplementary information is available in the online version of the paper.

Reprints and permissions information is available online at www.nature.com/reprints. Correspondence and requests for materials should be addressed to J.-L.B or M.W.

**Data availability**

All presented instrumental and modelling data of this study are available on the PANGAEA database with instructions about data format and necessary treatments.

## Author contributions

All authors contributed to the design of this study. Instrument layout and Picarro installation on Samoylov was done by J.-L.B., M.B., H.M., S.K., L.S., H.C.S.-L., and M.W. Isotope measurements and instrument maintenance were performed by J.-L.B., H.M. and M.W. The first manuscript draft was written by J.-L.B. and M.W., and all authors contributed to the discussion of results and the final article.

## Acknowledgements

This study has been funded by the AWI strategy fund project ISOARC.
We acknowledge the AARI, AWI and MPI SB RAS logistics for the invaluable support of the Russian-German LENA Expeditions and the Research Station Samoylov Island of IPGG SB RAS and its staff for support of our field work, in particular Andrei Astapov for help with the daily maintenance of the laser spectrometer.
We acknowledge the use of imagery from the NASA Worldview application (https://worldview.earthdata.nasa.gov), part of the NASA Earth Observing System Data and Information System (EOSDIS).

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
