# Peer review of "Moisture origin as a driver of temporal variabilities of the water vapour isotopic composition in the Lena River Delta, Siberia"

_Atmospheric Chemistry and Physics, 2019_

## Referee Comment (RC1) · Anonymous Referee #1 · 3 Feb 2020

General comments:

This paper uses a two year record of water vapor isotopes to help understand moisture sources and cycles of waters in the Siberian Arctic. This is an interesting paper that could be useful for both understanding patterns in the modern climate, but likely also has some applicability for paleo reconstructions (especially some of the moisture source isotope patterns). The seasonal differences in fractionation between phases are particularly interesting. The methods and study structure are largely sound and the interpretations make sense; with some moderate to minor changes this revised paper could be acceptable for further publication.

[Figure]

Specific comments: 74-78, 82-84: It would be helpful to have a distinct figure showing the location of the study site. I recognize the site can be seen on some of the other figures, but a location map figure would be helpful for orienting the study and the environmental variables in the region. 98: The sentence talks about "...parameters always measured above the snow cover", but there apparently is not always snow cover at the site (e.g., Figure 1). I think I know what you mean, but the description is a little confusing. Could you please clarify and/or rephrase? 110-111: It is stated that the container was heated, but at approximately what temperature? Was this the 50 âĎĊ of the inlet? Additionally, a photo or figure of the instrument set up might be helpful. 112: How was the inlet constantly heated at around 50 âĎĊ? 124-128: This is a little unclear: Are the humidity calibrated values relatively unchanged until 3 g/kg, then change logarithmically until 0.3 g/kg? 150-153: Approximately what percent of the total data set was removed because it lacked the full group of meteorological data? Is it possible that this is skewing any of your findings, such as the seasonal signals? 208-209: It is stated that it is difficult to investigate the impact of the change of each local moisture source as different surface cover changes (e.g., sea ice and snow cover) overlap. However, Figure 1 appears to show that sea ice cover starts to deteriorate in 6/2016 during a two to three month period in which there is no snow cover? Is it possible to use this period of time to try and disentangle the influence of sea ice versus snow cover? 268-270: Is the $\delta2H$ in reference to vapor? Is the R2 correlation coefficient between $\delta2H$ and $\delta18O$ really 1.0? Are there no differences? Perhaps I am misunderstanding something, but if $\delta2H$ and $\delta18O$ in vapor are changing exactly in time (e.g., 1.0 correlation coefficient) then wouldn't there not be in substantial differences in fractionation (e.g., switch from closer to equilibrium to more kinetic)? 278-282: It is stated that the minimum $\delta18Ov$ were observed in December and not February, which is interesting as February was the coldest and driest month. While this is technically true, February is not much colder and drier than December (0.6 âĎĊ colder and 0.04 g/kg). Are these differences really likely to explain the 3 per mil $\delta18Ov$ depletion difference in December, relative to February? 343-348: Some of the assumptions used for the theoretical vapor calculations are

stated, but it might be helpful to briefly state what model was used to make these theoretical calculations? Is there an error range associated with the theoretical vaues? 360: Again, a location figure would be useful to help understand the environment around the site and how seasonal changes could influence the isotope values.

361-363: It is stated that "incoming radiation might be insufficient to drive significant variations in evapotranspiration." Could these radiation variations be input into an ET model to see if this actually true? However, in light of the next sentence which talks about this explanation being inconsistent with actual ET data from Eddy-covariance tower observations, this is likely not needed. Just an option to consider.

Technical corrections:

Equation 2: The dot between 8 and $\delta$18O looks like a decimal place and not the multiplication symbol that it should be.

109: "(CRDS) has been installed..." This is passive, consider changing to "was installed"

116: add a space between "system(as described"

198-199: "measured snow depth indicates the permanent presence of a snow cover from"...Perhaps rephrase this as "permanent" snow cover appears to be in contrast from the September to June snow cover in this study.

Figure 1 Caption: Do liquid precipitation and snow cover depth also have 6 hour resolution?

268: Is the $\delta$2H in reference to vapor? .

---

## Referee Comment (RC2) · Anonymous Referee #2 · 2 Mar 2020

General comments: I believe that this paper is significant to see seasonal, synoptic variations of stable isotope ratios of vapor through 2-year collection of data. It is particular that it analyzes hydrological patterns by comparing not only with temperature and relative humidity, but also with wind directions, E-P and measured water vapor isotope ratio. Since characteristics of the study area are similar to those of polar regions, and it would be useful for studies of paleoclimate and modeling.

Specific comments:

135-138 Is there a comparison between IRMS and CRDS that measured stable isotope ratios of precipitation and vapor? Also, there is no clear explanation about the sampling

method of rain.

185 The word 'interannual' is used only once in this sentence throughout the paper. This word seems unnecessary if there is no reason.

215 Positions of decimal points should be consistent throughout the paper.

255 It is hard to see the variations stated in '185' by using this figure. You should use the background color or grids.

314-315 There should be more details about why wind speed was used to verify the safety of synoptic situation.

343 The word 'reason' seems more reasonable that 'origin'

343 It would be better to write the equation even though it is general.

347 It is hard to see variations like '255'.

372 Why is there no exchange reaction in other seasons?

448 Is only the coldest month attributed to paleoclimate? There should be an explanation.

451 There should be an explanation on Verkhoyansk mountains in the introduction.

478-480 I recommend to add Figure 8.

503-504 Isn't it high and low extrema during Yellow Shade period? The description of the figure should be modified.

483-495 The order of the figure should be corrected. (a) would be the first. You should adjust the paragraph or the figures.

4.3.2 influence of the moisture source

Can the red area be seen as the moisture source because the moments measured (evaporation – precipitation) and stable isotope ratios of vapor are same? Also, longdistance transportation doesn't mean isotope distillation. You should consider that initial isotope ratios of vapor would be different regarding to the ocean temperature.

Figure 5, 6, 9 Study area is denoted in green dots, but they are not clear. Azimuth should be indicated in the figure.

---

## Author Comment (AC1) · 28 May 2020

General comments:

*- RC1: This paper uses a two year record of water vapour isotopes to help understand moisture sources and cycles of waters in the Siberian Arctic. This is an interesting paper that could be useful for both understanding patterns in the modern climate, but likely also has some applicability for paleo reconstructions (especially some of the moisture source isotope patterns). The seasonal differences in fractionation between phases*

[Figure]

*are particularly interesting. The methods and study structure are largely sound and the interpretations make sense; with some moderate to minor changes this revised paper could be acceptable for further publication.*

Specific comments:

*- RC1: "74-78, 82-84: It would be helpful to have a distinct figure showing the location of the study site. I recognize the site can be seen on some of the other figures, but a location map figure would be helpful for orienting the study and the environmental variables in the region."*

> AC: Thank you for this suggestion. Such a map has been added into the corrected version of the manuscript.

*- RC1: "98: The sentence talks about "...parameters always measured above the snow cover", but there apparently is not always snow cover at the site (e.g., Figure 1). I think I know what you mean, but the description is a little confusing. Could you please clarify and/or rephrase¿'*

> AC: We understand that this could lead to misunderstanding. The sentence has been rephrased.

*- RC1: "110-111: It is stated that the container was heated, but at approximately what temperature? Was this the 50 °C of the inlet? Additionally, a photo or figure of the instrument set up might be helpful."*

> AC: The container heating is independent from the inlet tube heating. An electric heater situated in the container ensures the heating of the room. However, the room is not air conditioned, and the electric heater does not provide a stable temperature but avoids the temperature to drop below the freezing point during winter. This information has been added to the manuscript. A schematics of the analytical system have been added to the supplementary material.

*- RC1: "112: How was the inlet constantly heated at around 50 °C?"*

> AC: The inlet tube is insulated and the heating is carried out by a heating wire. This information has been added to the manuscript.

- *RC1: "124-128: This is a little unclear: Are the humidity calibrated values relatively unchanged until 3 g/kg, then change logarithmically until 0.3 g/kg?"*

> AC: The precision is almost stable for measurements above 3 g/kg. For measurements in drier air, the precision deteriorates logarithmically. The given estimates of the precision at 0.3 g/kg give an indication of the very dry conditions. The corresponding explanation has been modified in the manuscript.

- *RC1: "150-153: Approximately what percent of the total data set was removed because it lacked the full group of meteorological data? Is it possible that this is skewing any of your findings, such as the seasonal signals?"*

> AC: There is no lack of meteorological data but a lack of water vapour isotopic data. We remove 3.66% of the specific humidity data, 13.07% of the temperature data but none of the water vapour isotopic data. This method is only applied for the selection of synoptic events, as used in section 4.3.2 to evaluate the influence of the moisture sources based on the moisture source diagnostics. The interest of using this filtering method here is to avoid comparing a selection of synoptic events on different parameters which would not have been observed simultaneously. Thus, we might miss some extreme events which have been recorded on the temperature or specific humidity, but since our focus is to study the water vapour isotopic signal, we prefer selecting only the events for which the corresponding parameters have been measured. This method is not used for the other studies presented in this article, such as the computation of the seasonal cycle for example, as we wanted to use the maximum number of available data for this type of calculation. This has been clarified in the corrected version of the manuscript.

- *RC1: "208-209: It is stated that it is difficult to investigate the impact of the change of each local moisture source as different surface cover changes (e.g., sea ice and*

*snow cover) overlap. However, Figure 1 appears to show that sea ice cover starts to deteriorate in 6/2016 during a two to three month period in which there is no snow cover? Is it possible to use this period of time to try and disentangle the influence of sea ice versus snow cover?"*

> AC: The sea ice cover value that we show on this figure is the average sea ice cover over the surrounding 500km and thus covers part of the Laptev Sea but does not contain any information about the river ice in the Lena delta. What we noticed by looking at satellite pictures is that the river ice on the Lena delta disappears within a few days at the beginning of June, simultaneously with the snow cover melt. Then polynia open on the Laptev and large areas of fragmented ice remain over the sea, which require several months to completely melt and disappear. Hence, there is not a situation where the snow cover has disappeared but the sea ice completely remains, but it is a more complex evolution where part of the river and sea ice disappears simultaneously with snow, while some fragmented ice remains (mainly over the sea and not much in the Lena delta). There is no simple situation without snow but with a complete sea ice and river ice, or the opposite situation. Therefore, we believe that it is not possible to use this period to isolate a snow cover from a sea ice signal with the tools we are using in this study.

- *RC1: "268-270: Is the $\delta^2 H$ in reference to vapour? Is the $R^2$ correlation coefficient between $\delta^2 H$ and $\delta^{18} O$ really 1.0? Are there no differences? Perhaps I am misunderstanding something, but if $\delta^2 H$ and $\delta^{18} O$ in vapour are changing exactly in time (e.g., 1.0 correlation coefficient) then wouldn't there not be in substantial differences in fractionation (e.g., switch from closer to equilibrium to more kinetic)?"*

> AC: We are indeed referring to the vapour $\delta^2$H value. This omission has been corrected. Concerning $R^2$, we focus on the seasonal variation and this correlation coefficient between $\delta^2$H and $\delta^{18}O$ refers only to the monthly averaged values. The number of values is therefore very limited. There are however some differences when looking at higher temporal resolutions, which can be an indicator of some kinetic effects: for

example at a 6 hours resolution, $R^2$=0,98, with a slope of 7.15.

*- RC1: "278-282: It is stated that the minimum $\delta^{18}O_v$ were observed in December and not February, which is interesting as February was the coldest and driest month. While this is technically true, February is not much colder and drier than December (0.6 °C colder and 0.04 g/kg). Are these differences really likely to explain the 3 permil $\delta^{18}O_v$ depletion difference in December, relative to February?"*

> AC: We wanted to stress out that the differences in $\delta^{18}O_v$ between the two months (February and December) are in the opposite direction compared to the temperature and humidity values differences. However, it is true that the differences in humidity and temperatures between February and December are very limited. And we also believe that the differences in $\delta^{18}O_v$ values are too limited to be discussed, regarding their precision at very low humidity. Therefore, we decided to remove this remark from the manuscript.

*- RC1: "343-348: Some of the assumptions used for the theoretical vapour calculations are stated, but it might be helpful to briefly state what model was used to make these theoretical calculations? Is there an error range associated with the theoretical values?"*

> AC: The calculation is already detailed in the methods section 2.5: "We use the $\alpha_{eq}$ equilibrium fractionation coefficients between liquid (solid) and vapour for air temperatures above (below) the freezing point (Majoube, 1971a, 1971b; Merlivat and Nief, 1967). For each precipitation sample, with an isotopic ratio denoted RP, the theoretical isotopic ratio of vapour, denoted $R_v$, is given by $R_v = R_p/\alpha_{eq}$."

*- RC1: "360: Again, a location figure would be useful to help understand the environment around the site and how seasonal changes could influence the isotope values."*

> AC:A map showing the station environment has now been included in the manuscript (new figure 1), following the previous remark about lines 74-78, 82-84. Additionally,

aerial views of the station environment at different seasons has been included in the Supplementary materials (Supplementary figure 1) in order to highlight the seasonal surface cover changes.

*- RC1: "361-363: It is stated that "incoming radiation might be insufficient to drive significant variations in evapotranspiration." Could these radiation variations be input into an ET model to see if this actually true? However, in light of the next sentence which talks about this explanation being inconsistent with actual ET data from Eddy-covariance tower observations, this is likely not needed. Just an option to consider."*

> AC: We believe that the eddy covariance observations, performed a few hundred meters from our station are sufficient to prove that there are indeed diurnal variations of the evapotranspiration. Therefore, we did not perform such simulation. However, we have reformulated this paragraph in order to avoid misunderstandings that this could be a serious hypothesis.

Technical corrections:

*- RC1: "Equation 2: The dot between 8 and $\delta^{18}O$ looks like a decimal place and not the multiplication symbol that it should be."*

> AC: This has been corrected.

*- RC1: "109: '(CRDS) has been installed...' This is passive, consider changing to 'was installed'."*

> AC: We modified this sentence according to your suggestion.

*- RC1: "116: add a space between 'system(as described'."*

> AC: This typo has been corrected.

*- RC1: "198-199: "measured snow depth indicates the permanent presence of a snow cover from"... Perhaps rephrase this as "permanent" snow cover appears to be in contrast from the September to June snow cover in this study."*

> AC: What we meant is that the snow cover is present without interruption between September and June. We replaced the word "permanent" by "continuous", which we think is more appropriate.

- *RC1: "Figure 1 Caption: Do liquid precipitation and snow cover depth also have 6 hour resolution?"*

> AC: The liquid precipitation and snow cover depth are originally provided with a 30 minutes resolution and have been averaged at a 6 hours resolution for this figure. However, the precipitation isotopic values presented in this figure are daily averages of event based precipitation samples.

- *RC1: "268: Is the $\delta^2 H$ in reference to vapour?"*

> AC: We are indeed referring to the vapour value here. The information was missing in the text, but this is now included.

**Anonymous Referee 2**

General comments:

- *RC2: " I believe that this paper is significant to see seasonal, synoptic variations of stable isotope ratios of vapour through 2-year collection of data. It is particular that it analyses hydrological patterns by comparing not only with temperature and relative humidity, but also with wind directions, E-P and measured water vapour isotope ratio. Since characteristics of the study area are similar to those of polar regions, and it would be useful for studies of paleoclimate and modelling."*

Specific comments:

- *RC2: "135-138 Is there a comparison between IRMS and CRDS that measured stable isotope ratios of precipitation and vapour? Also, there is no clear explanation about the sampling method of rain."*

> AC: The IRMS measured precipitation samples while the CRDS continuously records

water vapour. There is no possibility to measure liquid samples using our CRDS instrument on-site, due to the system layout. We did not perform any sampling of the water vapour in the air, which would allow to measure the same water vapour with another technique, like IRMS. Hence, no samples have been measured on both the CRDS and the IRMS analysers so there is no direct comparison of the measurements of both analysers. Precipitation sampling was carried out after each rain and snowfall event. More details about the procedure for rain sampling has been included in the manuscript.

*- RC2: "185 The word 'interannual' is used only once in this sentence throughout the paper. This word seems unnecessary if there is no reason."*

> AC: Although this is not the main point of the paper, we do describe some interannual variations on the vapour and precipitation isotopic compositions between both winters (in section 3.1, lines 235-257). For this reason, we think it is reasonable to keep this word here.

*- RC2: "215 Positions of decimal points should be consistent throughout the paper."*

> AC: We have now corrected this aspect throughout the paper.

*- RC2: "255 It is hard to see the variations stated in '185' by using this figure. You should use the background colour or grids."*

> AC: Following the reviewer suggestion, we have added background grids to the figure.

*- RC2: "314-315 There should be more details about why wind speed was used to verify the safety of synoptic situation."*

> AC: As the synoptic variations are important compared to the diurnal cycles, we compute an average diurnal cycle for periods where the synoptic changes are low and where the observed air variations are supposedly due to local processes rather than large scale transport changes. We based our filtering method on the wind speed, with a threshold at 5 m s-1 over a duration of 24 hours, corresponding to the averaging

period. More details about this filtering have been provided in the corrected version of the manuscript.

*- RC2: "343 The word 'reason' seems more reasonable that 'origin'."*

> AC: This has been modified according to your suggestion

*- RC2: "343 It would be better to write the equation even though it is general."*

> AC: The calculation is already described in the method section.

*- RC2: "347 It is hard to see variations like '255'."*

> AC: We increased the size of the scatters to improve the readability of the figure.

*- RC2: "372 Why is there no exchange reaction in other seasons?"*

> AC: We did not intend to say that there couldn't be any exchange with the surface during other periods of the year. In the previous paragraph, we discuss about the other seasons and propose some hypotheses to explain the lack of diurnal variations of our isotopic signal during the other seasons, which are not necessarily explained by an absence of exchanges with the surface. As stated in this paragraph, variations of the evapotranspiration in summer linked with the incoming radiations in summer have been observed in summer (Helbig et al. 2013). However, our observations do not necessarily contradict these observations, for example if the evaporated flux has a similar isotopic signal than the atmospheric moisture. Here we try to explain the unique pattern of the significant diurnal cycle that we observe in May, compared to the rest of the year. The hypothesis that we propose involves a sublimation of the snow cover (deposited earlier in the winter season) driven by the diurnal variations of the insolation. This sublimation is probably not efficient enough earlier in the spring season as the insolation is still too low (and even lower during the polar night). It would stop later in spring the snow cover has completely melted. This section has been rephrased in order to clarify our hypotheses.

*- RC2: "448 Is only the coldest month attributed to paleoclimate? There should be an explanation."*

> AC: Local paleoclimate data retrieved in the Lena delta are based on ice wedges, which are formed from the snow deposited during winter and melting in spring. Therefore, the moisture of the coldest month, corresponding to the period where the snow is deposited, is the pertinent signal to study. This explanation has been included in the corrected version of the manuscript.

*- RC2: "451 There should be an explanation on Verkhoyansk mountains in the introduction."*

> AC: We have information about the location of the Verkhoyansk mountains in the Method section 2.1 describing the area of the study. The location of this mountain range has also been indicated on the new map of Figure 1.

*- RC2: "478-480 I recommend to add Figure 8."*

> AC: We refer here to an existing figure, so we added a reference to this figure in this paragraph.

*- RC2: "503-504 Isn't it high and low extrema during Yellow Shade period? The description of the figure should be modified."*

> AC: The legend of the figure has been rephrased.

*- RC2: "483-495 The order of the figure should be corrected. (a) would be the first. You should adjust the paragraph or the figures."*

> AC: The associated paragraphs has been reorganized in order to present the information in the same order as the figure.

*- RC2: "4.3.2 influence of the moisture source: Can the red area be seen as the moisture source because the moments measured (evaporation-precipitation) and stable isotope ratios of vapour are same? Also, long-distance transportation doesn't mean*

[Figure]

*isotope distillation. You should consider that initial isotope ratios of vapour would be different regarding to the ocean temperature."*

> AC: The maps presenting the "evaporation – precipitation" present an average situation for different types of synoptic events, based on the moisture source diagnostic only. There is no information regarding the isotopic values in these maps (apart from the selection of the synoptic events). Therefore, we do not know the isotopic composition of vapour at the source from these calculations. We do not pretend that it is possible to extrapolate our results to estimate the location of a moisture source only by similarity with the water vapour isotopic signal. The isotopic distillation is indeed not directly linked with the distance but associated with the decrease of temperatures. Therefore, we modified our text to point out that, for the low temperatures patterns associated with long distant transport from the oceanic sources, the important temperature variations between the moisture source and the location of our observations might drive a strong isotopic distillation.

*- RC2: "Figure 5, 6, 9 Study area is denoted in green dots, but they are not clear. Azimuth should be indicated in the figure."*

> AC: The marker size has been increased on these figures to improve clarity. We use a north pole Lambert azimuthal projection (azimuth is 90 °N) with a bounding latitude (tangent to the edge of the map) at 30 °N, centred on the longitude 0 °E. We added the information in the legends.